# Pregnancy-associated plasma protein-aa regulates endoplasmic reticulum–mitochondria associations

**Mroj Alassaf[1,2,3], Mary C Halloran[1,2,3]***

[1]Department of Integrative Biology, University of Wisconsin, Madison, United States; [2]Department of Neuroscience, University of Wisconsin, Madison, United States; [3]Neuroscience Training Program, University of Wisconsin, Madison, United States

**Abstract** Endoplasmic reticulum (ER) and mitochondria form close physical associations to facilitate calcium transfer, thereby regulating mitochondrial function. Neurons with high metabolic demands, such as sensory hair cells, are especially dependent on precisely regulated ER–mitochondria associations. We previously showed that the secreted metalloprotease pregnancy-associated plasma protein-aa (Pappaa) regulates mitochondrial function in zebrafish lateral line hair cells (Alassaf et al., 2019). Here, we show that *pappaa* mutant hair cells exhibit excessive and abnormally close ER–mitochondria associations, suggesting increased ER–mitochondria calcium transfer. *pappaa* mutant hair cells are more vulnerable to pharmacological induction of ER–calcium transfer. Additionally, *pappaa* mutant hair cells display ER stress and dysfunctional downstream processes of the ER–mitochondria axis including altered mitochondrial morphology and reduced autophagy. We further show that Pappaa influences ER–calcium transfer and autophagy via its ability to stimulate insulin-like growth factor-1 bioavailability. Together our results identify Pappaa as a novel regulator of the ER–mitochondria axis.

***For correspondence:**
mchalloran@wisc.edu

**Competing interests:** The authors declare that no competing interests exist.

## Introduction

Neuronal survival is critically dependent on proper mitochondrial function, which is regulated in part by close associations between mitochondria and the endoplasmic reticulum (ER). Despite the diverse pathologies of neurodegenerative diseases, increasing evidence suggests that a disrupted ER–mitochondria connection is a common underlying feature (*Area-Gomez et al., 2012*; *Bernard-Marissal et al., 2015*; *Calì et al., 2013*; *Hedskog et al., 2013*). Indeed, many of the cellular processes associated with neurodegeneration such as mitochondrial dysfunction, mitochondrial fragmentation, disrupted autophagy, and oxidative stress are regulated by the ER–mitochondria axis (*Csordás et al., 2018*; *Friedman et al., 2011*; *Rowland and Voeltz, 2012*).

Precise regulation of mitochondrial calcium levels is essential for mitochondrial metabolism and cell survival (*Cárdenas et al., 2010*). The association of mitochondria with ER facilitates efficient uptake of calcium by the mitochondria (*Krols et al., 2016*; *Rizzuto et al., 1998*). Calcium released from the ER forms highly concentrated calcium microdomains that are necessary to override the low affinity of the mitochondrial calcium uniporter (MCU), the primary mitochondrial calcium uptake channel (*Rizzuto et al., 2012*). However, overly close inter-organelle distances or excessive ER–mitochondria contact points can result in increased calcium transfer, which in turn can overstimulate mitochondrial bioenergetics leading to the production of pathological levels of the energy byproduct, reactive oxygen species (ROS). Accumulation of ROS can then lead to oxidative stress, fragmentation of the mitochondrial network, and cell death (*Adam-Vizi and Starkov, 2010*; *Brookes et al., 2004*; *Deniaud et al., 2008*; *Esterberg et al., 2014*; *Iqbal and Hood, 2014*; *Ježek et al., 2018*).

Thus, tight regulation of the distance and frequency of contacts between the ER and mitochondria is critical for proper mitochondrial calcium load. Previous studies investigating regulation of ER–mitochondria associations have focused on identifying proteins involved in tethering the ER to the mitochondria. However, the upstream molecular pathways that act to prevent excessive and pathologically tight ER–mitochondria associations, and the mechanisms by which these molecular pathways are regulated remain unknown.

To investigate these mechanisms, we are using zebrafish lateral line hair cells, which share molecular, functional, and morphological similarities with mammalian inner ear hair cells. Hair cells of the inner ear and lateral line system are specialized sensory neurons that transduce acoustic information and relay it to the central nervous system (*McPherson, 2018*). Sensory hair cells are an excellent model to investigate mechanisms regulating ER–mitochondria associations. Their high metabolic requirements make them particularly vulnerable to disruptions in mitochondrial calcium load (*Gonzalez-Gonzalez, 2017*). Indeed, excess mobilization of calcium from the ER to the mitochondria was recently shown to underlie aminoglycoside-induced hair cell death in the zebrafish lateral line (*Esterberg et al., 2014*). These findings highlight the importance of the ER–mitochondria connection in regulating hair cell survival. Given that hair cell death causes hearing loss (*Eggermont, 2017*), identifying molecular factors that regulate the ER–mitochondria connection may provide insight into potential therapeutic targets to combat ototoxin-induced hearing loss.

We showed previously that pregnancy-associated plasma protein-aa (Pappaa) regulates mitochondrial function in zebrafish lateral line hair cells (*Alassaf et al., 2019*). Pappaa is a locally secreted metalloprotease that regulates Insulin-like growth factor 1 (IGF1) signaling. IGF1 is sequestered extracellularly by IGF binding proteins (IGFBPs) that prevent it from binding to its receptor. Pappaa cleaves inhibitory IGFBs, thereby stimulating local IGF1 availability and signaling (*Hwa et al., 1999*). We found that loss of Pappaa, and the consequential reduction in IGF1 signaling, resulted in mitochondrial calcium overload, ROS buildup, and increased susceptibility to ROS-induced cell death (*Alassaf et al., 2019*). Moreover, other studies have shown that declining IGF1 levels correlate with conditions such as aging, neurodegenerative disorders, and obesity, in which the ER–mitochondria connection is known to be altered (*Arruda et al., 2014*; *Hedskog et al., 2013*; *Lee et al., 2018*; *Liu and Zhu, 2017*). However, a causative link between IGF1 signaling and ER–mitochondria associations has not been shown.

In this study, we reveal that Pappaa, an extracellular regulator of IGF1 signaling, is critical for regulation of the ER–mitochondria connection. We use electron microscopy (EM) to show that ER–mitochondria associations are closer and greater in number in *pappaa* mutant hair cells. Hair cells in *pappaa* mutants are more sensitive to pharmacological induction of ER-mediated calcium release and show changes in mitochondrial morphology and stunted autophagic response. Loss of Pappaa also results in ER stress and activation of the unfolded protein response. Together, our results suggest that Pappaa exerts its effect on hair cell survival by serving as a key regulator of the ER–mitochondria axis. Given the widespread roles for IGF1 receptors and the suppressive effects of IGFBPs, exogenous IGF1 treatment may not be an effective therapeutic approach for neurodegenerative diseases. A factor such as Pappaa that can locally activate IGF1 signaling may provide a more promising therapeutic target to prevent hearing loss.

## Results and discussion

### Pappaa influences ER–mitochondria associations

Zebrafish lateral line hair cells lie on the surface of the skin and are arranged into structures called neuromasts (*Figure 1A,B*). Each neuromast consists of a cluster of hair cells surrounded by the glia-like support cells. Our previous work showed that hair cells in *pappaa* mutants (hereafter referred to as *pappaa$^{p170}$*) are more susceptible to ROS-induced death and their mitochondria have increased calcium load (*Alassaf et al., 2019*). Therefore, we hypothesized that *pappaa$^{p170}$* mitochondria may have an increased frequency of ER-mitochondria associations. To test this, we used EM to visualize ER-mitochondria associations. We collected 80 nm thick sections along the apical-basal axis of anterior lateral line neuromasts of 5 days post fertilization (dpf) wild-type and *pappaa$^{p170}$* larvae

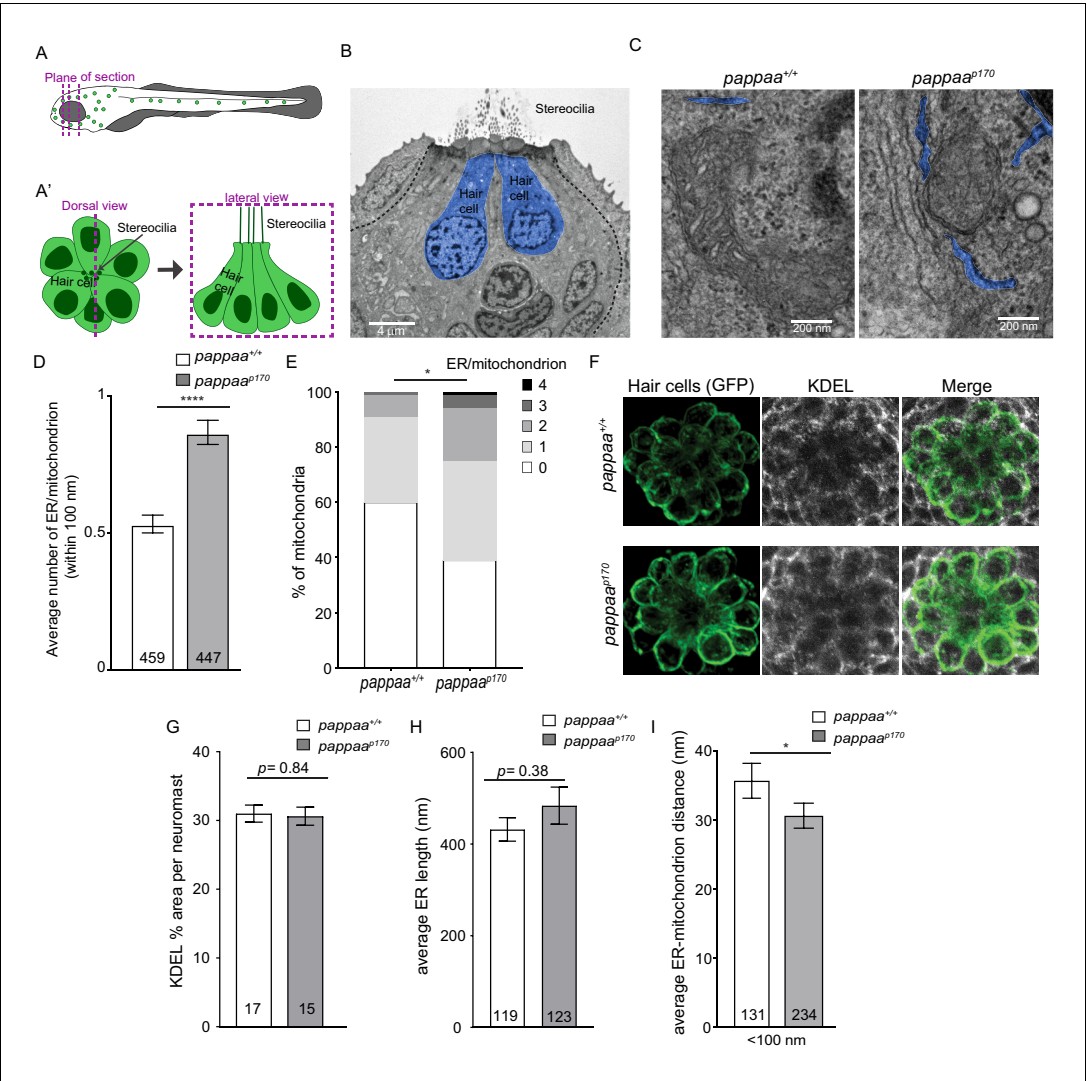

**Figure 1.** Pappaa regulates ER-mitochondria associations. (**A–A'**) Schematic of zebrafish lateral line hair cells. (**A**) The dotted lines represent EM plane of section. (**A'**) Schematic of a dorsal view (left) and lateral view (right) of a single neuromast. (**B**) Representative EM section of lateral line neuromast taken along the apical-basal axis of lateral line hair cells (blue) in 5 dpf larva. Scale bar = 4 µm. (**C**) Representative EM images of ER-mitochondria associations in wild-type and *pappaa* hair cells. ER is pseudo colored in blue. Scale bar = 200 nm. (**D**) Mean number of ER tubules within 100 nm of mitochondria. ****p<0.0001 t-test, Mann–Whitney correction. N = 459 mitochondria (wild type) and 447 mitochondria (*pappaa*^*p170*) collected from six larvae/genotype. Error bars = SEM. (**E**) Percentages of mitochondria associated with 0, 1, 2, 3, or 4 ER tubules. *p<0.05 chi-square test. N = 459 mitochondria (wild type) and 447 mitochondria (*pappaa*^*p170*) collected from six larvae/genotype. (**F**) KDEL immunolabeling in 5 dpf wild-type and *pappaa*^*p170* *brn3c:mGFP*-labeled hair cells. (**G**) Mean percentage of area covered by KDEL immunolabeling per neuromast. Unpaired t-test with Welch correction revealed no significant difference between groups p=0.84. N = 15–17 larvae/genotype (shown at base of bars), 1–3 neuromasts/larva. Total number of neuromasts included in the analysis = 35 (wild type) and 31 (*pappaa*^*p170*) neuromasts from two experiments. (**H**) Mean length of ER tubules. t-test with Mann–Whitney correction found no significant difference. N = 119 ER tubules (wild type) and 123 ER tubules (*pappaa*^*p170*) collected from six larvae/genotype. (**I**) Mean distance between the ER and mitochondria that are within 100 nm of each other. *p<0.05 t-test, Mann–Whitney correction. N = 131 ER-mitochondria associations (wild type) and 234 ER-mitochondria associations (*pappaa*^*p170*) collected from six larvae/genotype. Error bars=SEM.

The online version of this article includes the following source data and figure supplement(s) for figure 1:

**Source data 1.** Mean number of ER tubules/mitochondrion.
**Source data 2.** Percentages of mitochondria that are associated with 0, 1, 2, 3, or 4 ER tubules.
**Source data 3.** Mean percentage of area covered by KDEL immunolabeling.

*Figure 1 continued on next page*

*Figure 1 continued*

**Source data 4.** Mean ER tubule length.
**Source data 5.** Mean ER-mitochondria distance.
**Figure supplement 1.** Representative EM images of (A–A′) an efferent contact showing the post-synaptic ER
(arrow) and afferent (B–B′) contact identified by the synaptic ribbon (arrow).

(*Figure 1A,B*). Hair cells were identified based on their central location and darker cytoplasm as previously described (*Behra et al., 2009*; *Owens et al., 2007*; *Suli et al., 2016*). We quantified the number of ER tubules within 100 nm of mitochondria, the maximum distance for effective mitochondrial calcium uptake (*Csordás et al., 2018*). The ER was identified by its ribosome-rich membrane and elongated shape (blue profiles in *Figure 1C*). Because the ER at post-synaptic sites adjacent to efferent inputs has a distinct role in buffering high levels of post-synaptic calcium influx, and thus may impact the ER–mitochondria axis differently (*Moglie et al., 2018*), we did not include ER within 100 nm of the post-synaptic sites in our analysis. To identify the post-synaptic ER, we used the efferent terminal as a landmark, which presents as an electron light body filled with vesicles (*Figure 1— figure supplement 1A–A′*). We also did not include ER in EM sections with presynaptic terminals, identified by synaptic ribbons (*Figure 1—figure supplement 1B–B′*). We found that *pappaa^{p170}* hair cells had on average 34% more ER tubules in close association with each mitochondrion (*Figure 1C, D*). Furthermore, we quantified the percentage of mitochondria in close association with a given number of ER tubules. We found that 61% of mitochondria in *pappaa^{p170}* hair cells had one or more ER tubules in close association compared to 40% of wild-type hair cell mitochondria, and 25% of mutant mitochondria had two or more associated ER tubules compared with only 9% in wild type (*Figure 1C,E*). These results suggest that Pappaa regulates the frequency of ER–mitochondria associations.

We next asked whether the increase in the number of ER–mitochondria associations in *pappaa^{p170}* hair cells was due to an overabundance of ER. We used an antibody shown to specifically label KDEL (*Blanco-Sánchez et al., 2018*; *Blanco-Sánchez et al., 2014*), an ER C-terminal peptide retention signal (*Munro and Pelham, 1987*), to visualize the ER in the transgenic line *Tg(brn3c: mGFP)*, in which the lateral line hair cells are labeled with membrane-targeted GFP (*Figure 1F*). We measured the percentage of area occupied by KDEL immunofluorescence per neuromast using an approach previously described (*Blanco-Sánchez et al., 2014*). We found that the mean KDEL % area per neuromast was not different between wild-type and *pappaa^{p170}* larvae, suggesting there is no ER expansion in *pappaa^{p170}* hair cells (*Figure 1F–G*). As another means to assess ER abundance, we measured the length of ER tubules in our EM images. We found that there is no difference in the length of individual ER tubules (*Figure 1H*). Notably, the total number of ER tubules in this analysis did not significantly differ between wild type (119 tubules) and *pappaa^{p170}* (123 tubules) hair cells (*Figure 1H*). These results, in addition to our previous findings showing no difference in mitochondrial mass between wild-type and *pappaa^{p170}* hair cells (*Alassaf et al., 2019*), indicate that the increase in the number of ER-mitochondrion associations we see in *pappaa^{p170}* hair cells (*Figure 1D, E*) is not simply due to increased ER or mitochondrial mass, but instead reflects a specific effect on promoting close associations.

The distance between ER and mitochondria also influences the efficiency of mitochondrial calcium intake and must be tightly regulated (*Rizzuto et al., 2012*). Thus, we asked whether *pappaa^{p170}* hair cells also had tighter ER–mitochondria associations. We measured the distances between all ER and mitochondria within 100 nm of each other. We found that the ER-mitochondria distance was on average 5 nm shorter in *pappaa^{p170}* hair cells (*Figure 1C,I*). Taken together, our data suggest that Pappaa is important for preventing excessive and overly tight ER–mitochondria associations. Interestingly, another study used a synthetic linker to shorten the inter-organelle distance and showed a surge in mitochondria calcium load (*Csordás et al., 2006*), suggesting that the excessively tight connections we see in *pappaa^{p170}* hair cells could be causing their increased mitochondrial calcium load. Given the novelty of Pappaa's role in regulating ER-mitochondria associations, it will be interesting to identify the downstream molecular targets of Pappaa within the ER–mitochondria tethering complex.

## Pappaa loss sensitizes hair cells to increased ER–mitochondria calcium transfer

Given the increased ER–mitochondria associations in *pappaa*[p170] hair cells, we hypothesized that they experience excessive ER–mitochondria calcium transfer. If so, we would expect that *pappaa*[p170] hair cells would show increased vulnerability to pharmacological stimulation of ER-mitochondria calcium transfer. To address this question, we used thapsigargin, a sarco/endoplasmic reticulum $Ca^{2+}$-ATPase (SERCA) inhibitor, which prevents ER calcium influx and thereby allows for more available calcium in the cytosol to be taken up by the mitochondria (*Esterberg et al., 2014*; *Figure 2A*). A previous study used a transgenic zebrafish line with a mitochondria-targeted calcium indicator (mitoGCaMP3) to show that thapsigargin treatment caused increased mitochondrial calcium uptake in lateral line hair cells (*Esterberg et al., 2014*). We treated wild-type and *pappaa*[p170] larvae with thapsigargin and used the *Tg(brn3c:mGFP)* transgene to assess hair cell survival. We used a series of thapsigargin doses, either chronic exposure to low concentrations or acute exposure to high concentrations. We found that treatment for 24 hr with low doses of thapsigargin caused 11–21% more hair cell death in *pappaa*[p170] larvae compared to wild-type larvae (*Figure 2C*). Similarly, we found that 1 hr exposure to relatively high doses of thapsigargin at 5 dpf resulted in 13–22% more hair cell death in *pappaa*[p170] larvae (*Figure 2B,D*). These findings suggest that *pappaa*[p170] hair cells are more sensitive to fluctuations in calcium and further support the idea that *pappaa*[p170] hair cells have a disrupted ER–mitochondria connection that results in greater mitochondrial calcium uptake. However, it is difficult to discern whether the elevated levels of calcium in *pappaa*[p170] mitochondria are solely due to the altered ER-mitochondria associations. It is possible that *pappaa*[p170] hair cells experience increased ER calcium efflux and/or mitochondrial calcium uptake. Protein kinase B (Akt), which is a downstream effector of multiple signaling pathways including IGF1, has been shown to inhibit IP3Rs and reduce ER calcium release in HeLa cells (*Marchi et al., 2008*). If Akt has a similar role in hair cells, it is possible that Pappaa-IGF1 signaling may normally act to stimulate Akt-induced suppression of IP3Rs, preventing excessive ER calcium release. Further investigation into the activity of ER and mitochondria associated calcium channels in *pappaa*[p170] hair cells is needed to answer this question.

We sought to define the molecular pathway by which Pappaa regulates ER-mitochondria calcium transfer. Pappaa is a secreted metalloprotease and a known regulator of IGF1 signaling. Through its proteolytic activity, Pappaa cleaves the inhibitory IGFBPs, freeing IGF to bind to its receptor and thereby increasing IGF bioavailability and signaling (*Boldt and Conover, 2007*). To determine whether Pappaa regulates ER–mitochondria calcium transfer by stimulating IGF1 signaling, we took two approaches. First, we reasoned that if IGF1 signaling plays a role in regulating ER–mitochondria calcium transfer, then attenuating IGF1 signaling would result in increased sensitivity of wild-type hair cells to thapsigargin. Second, we reasoned that pharmacological stimulation of IGF1 bioavailability in *pappaa*[p170] mutants would improve hair cell survival in response to thapsigargin. To attenuate IGF1 signaling, we used a transgenic line with an inducible heat shock promoter that drives the expression of a dominant negative IGF1 receptor [*Tg(hsp70:dnIGF1Ra-GFP)*] (*Kamei et al., 2011*). Given that we previously showed IGF1 signaling acts post-developmentally to support hair cell survival (*Alassaf et al., 2019*), we induced *dnIGF1R* expression on day four and exposed the larvae to thapsigargin for 1 hr. Notably, heat shock alone in wild-type or *Tg(hsp70:dnIGF1Ra-GFP)* larvae did not alter hair cell survival (*Figure 2E*). However, expression of *dnIGF1R* did exacerbate hair cell loss in response to thapsigargin. Interestingly, 10 μM thapsigargin did not kill wild-type hair cells; however, it did cause hair cell death in *dnIGF1Ra-GFP* expressing larvae (*Figure 2E*), similar to *pappaa*[p170] larvae. This suggests that IGF1R signaling attenuates ER–mitochondria calcium transfer. To stimulate IGF1 bioavailability in *pappaa*[p170] mutants, we treated larvae with NBI-31772, an IGFFBP inhibitor, for 24 hr and then exposed the larvae to a 1 hr treatment with thapsigargin. We found that treatment with NBI-31772 significantly improved hair cell survival in *pappaa*[p170] larvae (*Figure 2F*). These data support the idea that Pappaa regulates ER-mitochondria calcium transfer via its ability to increase the bioavailability of IGF1.

Pappaa is not expressed by the hair cells themselves, rather it is expressed by the surrounding support cells of the neuromast (*Alassaf et al., 2019*). Because Pappaa is a secreted protein, it could potentially affect IGF signaling in the hair cells, support cells, or both. To ask which cells show altered IGF1 signaling in *pappaa*[p170] mutants, we used an antibody against phosphorylated IGF1R

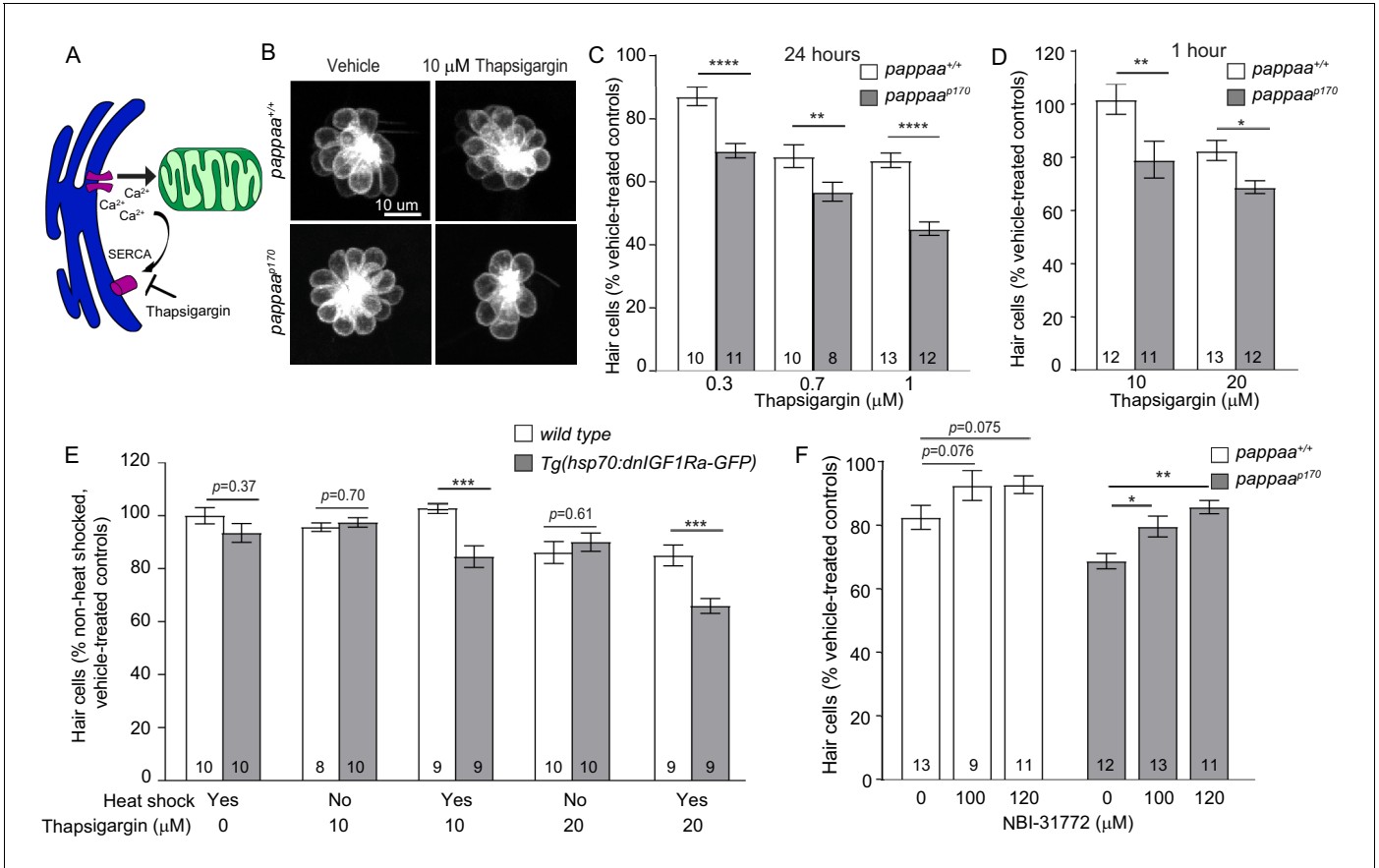

**Figure 2.** *pappaa^{p170}* hair cells are more sensitive to disruption in ER–mitochondria calcium signaling. (**A**) Thapsigargin increases calcium concentration at the ER-mitochondria junction by blocking the SERCA pump and inhibiting calcium uptake by the ER. (**B**) Representative images of *brn3c:mGFP*-labeled hair cells from vehicle or 10 μM thapsigargin-treated larvae. (**C**) Mean percentage of surviving hair cells following 24 hr treatments with thapsigargin starting at 4 dpf. To calculate hair cell survival percentage, hair cell number post-drug treatment was normalized to mean hair cell number in vehicle treated larvae of the same genotype. **p<0.01, ****p<0.0001, two-way ANOVA, Holm–Sidak post-test. N = 8–13 larvae per group (shown at base of bars), three neuromasts/larva were analyzed. Total number of neuromasts included in the analysis = 24 (wild type; vehicle-treated), 24 (*pappaa^{p170}*; vehicle-treated), 30 (wild type; 0.3 μM thapsigargin), 33 (*pappaa^{p170}*; 0.3 μM thapsigargin), 30 (wild type; 0.7 μM thapsigargin), 24 (*pappaa^{p170}*; 0.7 μM thapsigargin), 39 (wild type; 1 μM thapsigargin), 36 (*pappaa^{p170}*; 1 μM thapsigargin). (**D**) Mean percentage of surviving hair cells following 1 hr treatment with thapsigargin at 5 dpf. *p<0.05, **p<0.01, two-way ANOVA, Holm–Sidak post-test. N = 8–13 larvae per group (shown at base of bars), three neuromasts/larva from two experiments were analyzed. Total number of neuromasts included in the analysis = 60 (wild type; vehicle-treated), 60 (*pappaa^{p170}*; vehicle-treated), 36 (wild type; 10 μM thapsigargin), 33 (*pappaa^{p170}*; 10 μM thapsigargin), 39 (wild type; 20 μM thapsigargin), 36 (*pappaa^{p170}*; 20 μM thapsigargin). (**E**) Mean percentage of surviving hair cells following induction of *dnIGF1R* expression. To calculate hair cell survival percentage, hair cell number after 1 hr treatments with thapsigargin was normalized to mean hair cell number in non-heat-shocked, vehicle-treated larvae of the same genotype. ***p<0.001 two-way ANOVA, Holm–Sidak post-test. N = 8–10 larvae per group (shown at base of bars), three neuromasts per larva. Total number of neuromasts included in the analysis = 30 (wild type; non-heat-shocked, vehicle-treated), 27 (*dnIGF1Ra*; non-heat-shocked, vehicle-treated), 30 (wild type; heat-shocked, vehicle-treated), 30 (*dnIGF1Ra*; heat-shocked, vehicle-treated), 24 (wild type; non-heat-shocked, 10 μM thapsigargin), 30 (*dnIGF1Ra*; non-heat-shocked, 10 μM thapsigargin), 27 (wild type; heat-shocked, 10 μM thapsigargin), 27 (*dnIGF1Ra*; heat-shocked, 10 μM thapsigargin), 30 (wild type; non heat-shocked, 20 μM thapsigargin), 30 (*dnIGF1Ra*; non-heat-shocked, 20 μM thapsigargin), 27 (wild type; heat-shocked, 20 μM thapsigargin), 27 (*dnIGF1Ra*; heat-shocked, 20 μM thapsigargin). (**F**) Mean percentage of surviving hair cells following co-treatment with NBI-31772 and 20 μM thapsigargin. To calculate hair cell survival percentage, hair cell counts after treatment were normalized to hair cell number in vehicle treated larvae of the same genotype. *p<0.05, **p<0.01. Two-way ANOVA, Holm–Sidak post-test. N = 9–13 larvae per group (shown at base of bars), three neuromasts per larva. Total number of neuromasts included in the analysis = 24 (wild type; vehicle-treated), 24 (*pappaa^{p170}*; vehicle-treated), 39 (wild type; 20 μM thapsigargin), 36 (*pappaa^{p170}*; 20 μM thapsigargin), 27 (wild type; 20 μM thapsigargin+ 100 μM NBI−31772), 39 (*pappaa^{p170}*; 20 μM thapsigargin + 100 μM NBI−31772), 33 (wild type; 20 μM thapsigargin + 120 μM NBI−31772), and 33 (*pappaa^{p170}*; 20 μM Thapsigargin + 120 μM NBI−31772).

The online version of this article includes the following source data and figure supplement(s) for figure 2:

**Source data 1.** Hair cell survival following 24 hr treatment with thapsigargin in wild-type and *pappaa^{p170}* larvae.

**Source data 2.** Hair cell survival following 1 hr treatment with thapsigargin in wild-type and *pappaa^{p170}* larvae.

*Figure 2 continued on next page*

*Figure 2 continued*

**Source data 3.** Hair cell survival following induction of *dnIGF1R* expression.
**Source data 4.** Hair cell survival following co-treatment with thapsigargin and NBI-31772 in wild-type and *pappaa*[p170] larvae.
**Figure supplement 1.** Anti-pIGF1R immunolabeling.
**Figure supplement 1—source data 1.** Mean pIGF1R fluorescence in wild-type and *pappaa*[p170] hair cells.
**Figure supplement 1—source data 2.** Mean pIGF1R fluorescence in wild-type and *pappaa*[p170] support cells.

(pIGF1R) that was previously validated using an IGF1R inhibitor (*Chablais and Jazwinska, 2010*). IGF1R undergoes autophosphorylation when it binds to IGF1, thereby activating its signaling cascade (*Feldman et al., 1997*). We quantified the fluorescent intensity of pIGF1R in wild-type and *pappaa*[p170] neuromasts. We found that loss of Pappaa caused a substantial reduction in pIGF1R fluorescence in the hair cells (*Figure 2—figure supplement 1A–B*), suggesting that Pappaa secreted by support cells can impact IGF1R signaling in the neighboring hair cells. Interestingly, the support cells in *pappaa*[p170] mutants also showed reduced pIGF1R fluorescence (*Figure 2—figure supplement 1A,C*), suggesting that Pappaa can have an autocrine effect on support cells. It is possible that hair cells are also affected indirectly by reduced IGF signaling in support cells. In support of this idea, activation of phosphoinositide 3-kinase (PI3K), which is downstream of IGF1R, in the support cells of mammalian cochlea resulted in enhanced survival of hair cells against cisplatin, a chemotherapy agent, suggesting that IGF1 signaling in support cells could influence hair cells. However, the mechanism by which PI3K signaling in support cells ultimately affects cochlear hair cells is not fully understood (*Jadali et al., 2017*).

## Pappaa regulates mitochondrial morphology

Mitochondria undergo constant fission and fusion as a form of quality control. Damaged mitochondria are excised from the mitochondrial network through the process of fission and cleared by autophagy, while the remaining reusable pool of mitochondria fuse back together (*Ni et al., 2015*). Thus, a balance between mitochondrial fission and fusion must be achieved to maintain cellular homeostasis (*Liesa et al., 2009*). Under pathological conditions, such as oxidative stress in which there is an excess of damaged mitochondria, the balance between fission and fusion tips toward fission. Excessive fission can lead to fragmentation of the mitochondrial network and ultimately to cell death. Recently, it was shown that mitochondrial fission occurs at the site of ER-mitochondria contacts. ER tubules were found to wrap around mitochondria and mark the site for fission (*Friedman et al., 2011*). Given that *pappaa*[p170] hair cells experience oxidative stress, have dysfunctional mitochondria (*Alassaf et al., 2019*), and exhibit increased ER-mitochondria associations, we hypothesized that *pappaa*[p170] hair cells experience increased fission. Mitochondrial morphology can be predictive of fission and fusion events. Small, round, and clustered mitochondria are often indicative of fission events, whereas large and branched mitochondria are indicative of fusion events. To test our hypothesis, we used EM and measured several parameters of mitochondrial morphology that are typically used as a readout of mitochondrial fragmentation (*Wiemerslage and Lee, 2016*). These parameters include the area, perimeter, aspect ratio (major axis/minor axis; a measure of elongation), circularity ($4\pi \times$ area/perimeter$^2$), and network interconnectivity (area/perimeter) (*Wiemerslage and Lee, 2016*). We found that *pappaa*[p170] mitochondria had a smaller area, perimeter, and aspect ratio than wild-type mitochondria. Moreover, they were more circular and showed reduced network interconnectivity, suggesting excessive mitochondrial fission and possible network fragmentation (*Figure 3A–F*). To further investigate mitochondrial integrity in whole cells, we labeled hair cells with the vital mitochondrial dye Mitotracker. Using confocal microscopy, we acquired z-stacks of neuromast hair cells. Mitochondria in *pappaa*[p170] hair cells appeared less connected and more rounded compared to wild type (*Figure 3G*, *Videos 1* and *2*). Analysis of mitochondrial circularity using ImageJ (*Figure 3H*) revealed results similar to the EM data, supporting the idea that *pappaa*[p170] mitochondria are fragmented. Taken together, our data suggest that Pappaa regulates mitochondrial fission. Our findings are consistent with a previous study showing that treatment with IGF1 maintained mitochondrial integrity in denervated muscle cells by promoting mitochondrial fusion and preventing mitochondrial fission (*Ding et al., 2017*).

## Pappaa regulates neomycin-induced autophagy

We next asked if other processes known to be regulated by the ER–mitochondria axis are disrupted in *pappaa*^(p170) hair cells. Multiple studies suggest that autophagy, a bulk degradation process, is strongly influenced by the ER–mitochondria axis. Pharmacological and genetic manipulations that tighten or increase ER–mitochondria associations result in reduced autophagosome formation (*Gomez-Suaga et al., 2017a*; *Gomez-Suaga et al., 2017b*; *Sano et al., 2012*). Therefore, we asked whether *pappaa*^(p170) hair cells have an attenuated autophagic response. Interestingly, neomycin, to which *pappaa*^(p170) hair cells show hypersensitivity (*Alassaf et al., 2019*), was shown to be sequestered into autophagosomes upon entry into hair cells (*Hailey et al., 2017*; *He et al., 2017*). These authors also showed that blocking autophagy exacerbates neomycin-induced damage presumably due to the heightened exposure of vital intracellular components to neomycin. Thus, neomycin uptake into autophagosomes can be used as an indicator of the robustness of the autophagic response. To test whether *pappaa*^(p170) hair cells show a dampened autophagic response to neomycin treatment, we used a fluorescently tagged neomycin (Neomycin-Texas Red). Upon cell entry through the mechanotransduction (MET) channels (*Alharazneh et al., 2011*; *Hailey et al., 2017*; *Kroese et al., 1989*), neomycin-Texas Red appears as a diffused pool in the cytosol. Within minutes, neomycin gets captured by autophagosomes and high fluorescent intensity puncta begin to appear (*Hailey et al., 2017*; *Figure 4* and *Video 3*). We counted the number of puncta within visually accessible neuromast hair cells in wild-type and *pappaa*^(p170) larvae. We found that wild-type hair cells had 300% more neomycin-Texas red puncta. (*Figure 4B,C* and *Video 4*), indicating that neomycin-induced autophagy was attenuated in *pappaa*^(p170) hair cells. This reduction in autophagy may allow neomycin to accumulate in the cytosol, causing more damage to intracellular compartments. To test the possibility that the increased neomycin-induced hair cell death in *pappaa*^(p170) larvae (*Alassaf et al., 2019*) is the result of enhanced entry of neomycin through the MET, we measured the change in fluorescent intensity of neomycin-Texas Red (ΔF/F0) in individual hair cells at 2.5, 4.5, and 6.5 min post neomycin-Texas Red exposure. We found that *pappaa*^(p170) hair cells displayed similar neomycin-Texas Red ΔF/F0 to wild-type hair cells at every time point (*Figure 4D*). To determine whether *pappaa*^(p170) hair cells experience overall increased neomycin-Texas Red loading, we looked at the maximum ΔF/F0 and found no difference between *pappaa*^(p170) and wild-type hair cells (*Figure 4E*). This is consistent with our previous finding showing that MET channel-mediated entry of the fluorescent dye FM1-43 was not altered in *pappaa*^(p170) hair cells (*Alassaf et al., 2019*).

We next asked whether Pappaa regulates autophagy by increasing the availability of IGF1. To answer this, we treated *pappaa*^(p170) larvae with 120 µM NBI-31772 for 24 hr starting at 4 dpf before exposure to neomycin-Texas red at 5 dpf. Compared to vehicle treated *pappaa*^(p170) controls, which show an average of 1 neomycin-Texas Red punctate/hair cell, treatment with NBI-31772 resulted in a threefold increase in the number of neomycin-Texas Red puncta (*Figure 4F–G*), which is similar to wild-type levels (*Figure 4C*). Together, these findings suggest that Pappaa regulates neomycin-induced autophagy by increasing the availability of IGF1.

## Pappaa loss causes ER stress

ER stress can be a potential cause of mitochondria dysfunction. In addition to acting as the largest calcium reserve in the cell, the ER is the primary hub for protein processing. Newly synthesized proteins enter the ER for chaperone-assisted folding (*Gregersen et al., 2006*). Improper or insufficient folding results in the accumulation of unfolded proteins, which causes ER stress and increased calcium efflux (*Houck et al., 2012*). This efflux results in an increase in highly concentrated calcium 'hot spots' that act as arresting signals for mitochondria, tethering the mitochondria to the ER and thereby increasing the frequency and tightness of ER–mitochondria associations (*Bravo et al., 2012*; *Bravo et al., 2011*). To ask whether *pappaa*^(p170) hair cells experience ER stress, we FACsorted hair cells from wild-type and *pappaa*^(p170) *Tg(brn3c:mGFP)* larvae and evaluated gene expression of the unfolded protein response (UPR) by RT-qPCR. The UPR consists of three ER transmembrane receptors that act as sensors for unfolded proteins, IRE1, ATF4, and PERK (*Figure 5A*). Normally, Bip, an ER-resident chaperone, is bound to the UPR receptors. When unfolded proteins begin to accumulate, Bip dissociates from the UPR receptors to assist in protein folding. The uncoupling of Bip activates the UPR signaling cascade signifying ER stress. During the early phase of ER stress, the UPR promotes cell survival by improving the folding capacity of the ER through the upregulation of pro-

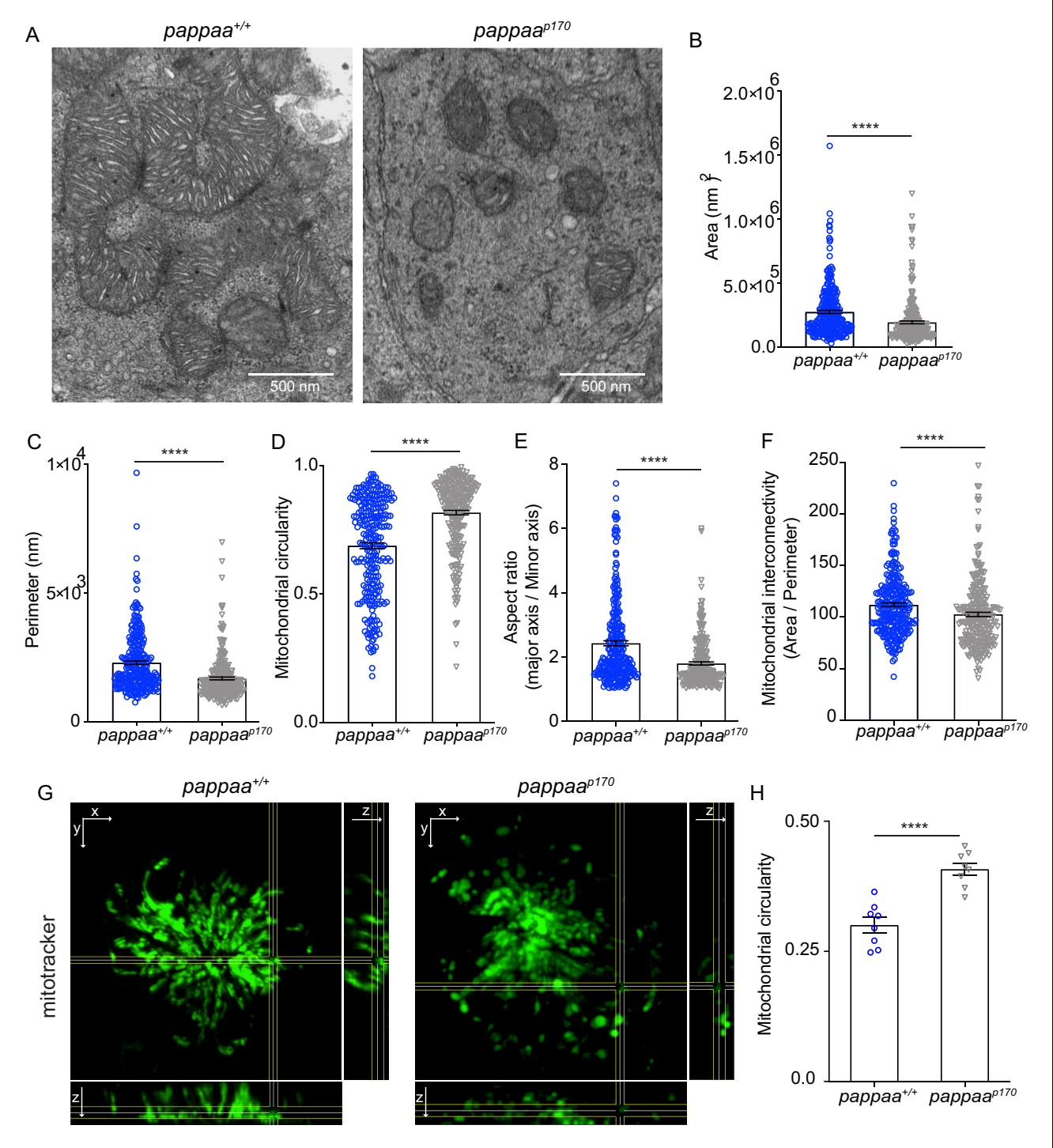

**Figure 3.** Pappaa loss causes mitochondrial fragmentation. (**A**) Representative EM images of mitochondria in lateral line hair cells in 5 dpf wild-type and *pappaa^p170* larvae. (**B–F**) Mean mitochondrial (**B**) area, (**C**) perimeter, (**D**) circularity, (**E**) aspect ratio, and (**F**) interconnectivity in 5 dpf wild-type and *pappaa^p170* lateral line hair cells. ****p<0.0001 t-test, Mann–Whitney correction. N = 272 mitochondria (wild type) and 262 mitochondria (*pappaa^p170*) collected from six larvae/genotype. (**G**) Representative images of 5 dpf wild-type and *pappaa^p170* lateral line hair cells loaded with the vital mitochondrial dye, Mitotracker. Images are maximum intensity projection through neuromast in xy view, with cross sections of yz plane shown at right and xz plane shown at bottom. (**H**) Mean mitochondrial circularity measured from Z-stack max intensity projections of wild-type and *pappaa^p170* lateral line hair cells. ****p<0.0001 t-test, Welch correction. N = 8 larvae per group (shown at base of bars), one neuromast/ larva. Error bars=SEM. See *Videos 1* and *2*.

The online version of this article includes the following source data for figure 3:

**Source data 1.** Mitochondrial area in wild-type and *pappaa^p170* lateral line hair cells.

**Source data 2.** Mitochondrial perimeter in wild-type and *pappaa^p170* lateral line hair cells.
**Source data 3.** Mitochondrial circularity in wild-type and *pappaa^p170* lateral line hair cells.
**Source data 4.** Mitochondrial aspect ratio in wild-type and *pappaa^p170* lateral line hair cells.
**Source data 5.** Mitochondrial interconnectivity in wild-type and *pappaa^p170* lateral line hair cells.
**Source data 6.** Mitochondrial circularity in wild-type and *pappaa^p170* lateral line hair cells measured by mitotracker.

survival factors including *bip*, *atf4*, and the splicing of *xbp1*. However, a switch from an adaptive to a pro-apoptotic UPR occurs during the late phase of ER stress, in which *chop*, a pro-apoptotic transcription factor, is upregulated (*Fu et al., 2015*; *Oslowski and Urano, 2011*). Our analysis revealed that components of the 'adaptive' UPR pathway (*bip*, *atf4*, and *spliced-xbp1*) were upregulated in *pappaa^p170* hair cells (*Figure 5B*). Notably, the 'pro-apoptotic' factor, *chop*, was not differentially expressed in *pappaa^p170* hair cells. These results suggest that *pappaa^p170* hair cells experience early ER stress but not the later, pro-apoptotic response. To further investigate whether *pappaa^p170* hair cells are in the process of apoptosis, we performed terminal deoxynucleotidyl transferase dUTP nick-end labeling (TUNEL) and used phalloidin as a counterstain to identify hair cells. As a positive control, we treated wild-type larvae with neomycin, a known apoptosis-inducing drug, for 30 min prior to fixation, as previously described (*Goodman and Zallocchi, 2017*; *Figure 5C*). In all neuromasts analyzed (head neuromasts in 10 larvae/group), we observed multiple TUNEL-positive hair cells in the positive controls. In contrast, in both untreated wild-type and *pappaa^p170* larvae, we did not observe any TUNEL-positive neuromasts, which is consistent with the lack of spontaneous hair cell death in *pappaa^p170* larvae (*Alassaf et al., 2019*). Together with our finding that *pappaa^p170* hair cells do not exhibit ER expansion (*Figure 1C,F*), an event associated with advanced ER stress as a result of extreme accumulation of misfolded proteins in the ER (*Chavez-Valdez et al., 2016*; *Dorner et al., 1989*; *Welihinda et al., 1999*), these data support the idea that the ER in *pappaa^p170* hair cells is in the early stages of stress. Interestingly, another study using a zebrafish model of Usher syndrome, a genetic disorder that can cause hearing loss (*Blanco-Sánchez et al., 2014*), found that while Usher mutant larvae did not show spontaneous hair cell death, the hair cells were TUNEL positive and showed ER expansion, suggesting that Usher mutants experience more advanced stages of ER stress than *pappaa^p170* mutants.

To ask whether *pappaa^p170* hair cells are more vulnerable to pharmacological induction of protein unfolding given their already active UPR, we treated 5 dpf wild-type and *pappaa^p170* Tg(brn3c: mGFP) larvae with tunicamycin, an inhibitor of glycoprotein biosynthesis (*Oslowski and Urano, 2011*), for 24 hr, and analyzed hair cell survival. *pappaa^p170* larvae showed 20% less hair cell survival compared to wild type, suggesting that they are closer to the cytotoxic threshold of ER stress (*Figure 5D*). Together, these findings suggest that Pappaa is essential for ER homeostasis.

## Conclusion

The ER and mitochondria are physically and functionally linked together to facilitate a range of cellular processes essential for neuron survival. Disruptions to this connection can cause damage to vital cellular components and is found to occur in many neurodegenerative diseases. To identify potential targets for therapeutic interventions, we must define and characterize the genetic and molecular mechanisms regulating the ER–mitochondria axis. Here, we describe a novel role for Pappaa, a local stimulator of IGF1 signaling, in regulating the ER–mitochondria axis and its essential downstream processes. We previously showed that mitochondria in *pappaa* mutant hair cells are dysfunctional, resulting in oxidative stress (*Alassaf et al., 2019*). Here we provide additional mechanistic insight into Pappaa's role

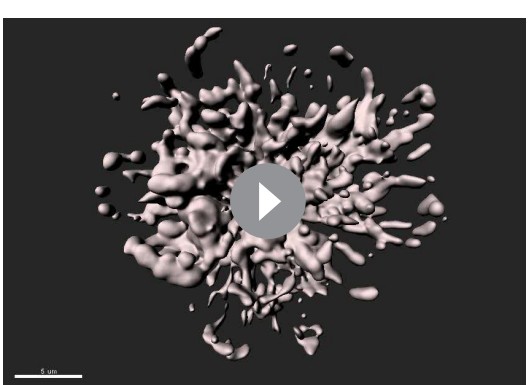

**Video 1.** 3D rendering of mitotracker labeling in wild-type lateral line hair cells. This video was constructed from a confocal z-stack of wild-type lateral line hair cells loaded with mitotracker.
https://elifesciences.org/articles/59687#video1

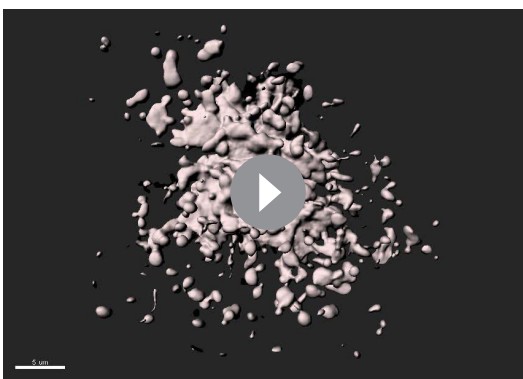

**Video 2.** 3D rendering of mitotracker labeling in *pappaa^{p170}* lateral line hair cells. This video was constructed from a z-stack of *pappaa^{p170}* lateral line hair cells loaded with mitotracker.
https://elifesciences.org/articles/59687#video2

in hair cell survival. We show that Pappaa is essential for the function of ER and its relationship to mitochondria in hair cells. Hair cell function and activity are influenced by the surrounding support cells as well as the efferent and afferent neurons that innervate hair cells (*Babola et al., 2020*; *Faucherre et al., 2009*; *Pichler and Lagnado, 2020*). A caveat to our pharmacological experiments is that drug treatments will affect all of these cell types, and some effects on hair cells may be indirect. Nonetheless, we show that the ER stress, excessive and abnormally tight ER–mitochondria associations, defects in mitochondria morphology and reduced autophagic activity in *pappaa^{p170}* mutants occur in the hair cells. A challenge to determining the initial site of dysfunction within *pappaa^{p170}* hair cells is the functional coupling of the ER and mitochondria. ER-mediated calcium release acts as a key regulator of mitochondrial bioenergetics, and thereby ROS production (*Görlach et al., 2015*). In turn, mitochondria-generated ROS can stimulate the activity of ER-associated calcium channels causing further calcium release (*Brookes et al., 2004*; *Chaudhari et al., 2014*; *Görlach et al., 2015*). Given that many ER-resident molecular chaperones are calcium dependent, ER stress can be both a cause and a consequence of mitochondrial dysfunction (*Zhu and Lee, 2015*). While it is difficult to definitively determine which organelle is affected first when the ER–mitochondria axis is compromised, several of our findings support an ER-driven dysfunction in *pappaa^{p170}* mutants. *Bravo et al., 2011* showed that during early ER stress, increased ER–mitochondria coupling occurs to enhance mitochondrial bioenergetics. *pappaa^{p170}* hair cells display all the telltale signs of early ER stress, including activation of the adaptive UPR branch (*Figure 5B*), heightened mitochondrial bioenergetics evident by increased calcium load, ROS, and hyperpolarization (*Alassaf et al., 2019*). Furthermore, *Esterberg et al., 2014* characterized the timeline of subcellular events following ER calcium release in hair cells. They showed that calcium released from the ER is quickly taken up by the mitochondria and initially causes mitochondria hyperpolarization suggesting increased bioenergetics. However, once a certain threshold is crossed, depolarization of the mitochondria occurs, which is followed by permeabilization of the outer membrane and eventually cell death, hence this step is often referred to as 'the point of no return'. *pappaa^{p170}* mitochondria are hyperpolarized (*Alassaf et al., 2019*), which places them a step after ER–calcium release and before depolarization of the outer mitochondrial membrane and hair cell death. Indeed, *pappaa^{p170}* hair cells do not spontaneously degenerate (*Figure 5C*). Although it is difficult to determine with absolute certainty, we speculate that ER stress may precede mitochondrial dysfunction in *pappaa^{p170}* hair cells. Precise temporal dissection is needed to identify Pappaa's primary subcellular target, whether the mitochondria, ER, or both.

IGF1 emerges as a promising therapeutic candidate due to its known role in influencing the function of the ER and the mitochondria. Studies using exogenous IGF1 treatment showed it can alleviate pharmacologically induced ER stress in cancer cells (*Novosyadlyy et al., 2008*) or attenuate mitochondria-generated ROS in cirrhosis or aging disease models (*Castilla-Cortazar et al., 1997*; *García-Fernández et al., 2008*; *Novosyadlyy et al., 2008*; *Sádaba et al., 2016*), supporting the possibility that IGF1 signaling may play a role in regulating the ER-mitochondria axis. However, exogenous IGF1 treatment presents its own set of challenges for developing an IGF-based therapy. IGF1 signaling has a broad cellular impact and ubiquitous overexpression could affect many systems. Additionally, the suppressive effects of IGFBPs on IGF1 may render this approach inefficient. Indeed, patients with neurodegenerative disorders have not shown significant improvement following systemic IGF1 administration. These disappointing outcomes are thought to be due to the suppressive effects of IGFBPs on IGF1 bioavailability (*Raoul and Aebischer, 2004*; *Sakowski et al., 2009*). This was evident in patients with amyotrophic lateral sclerosis, a progressive motor neuron disease (*Bowling et al., 1993*; *Shaw et al., 1995*), in which total IGF1 levels appeared to be normal, whereas

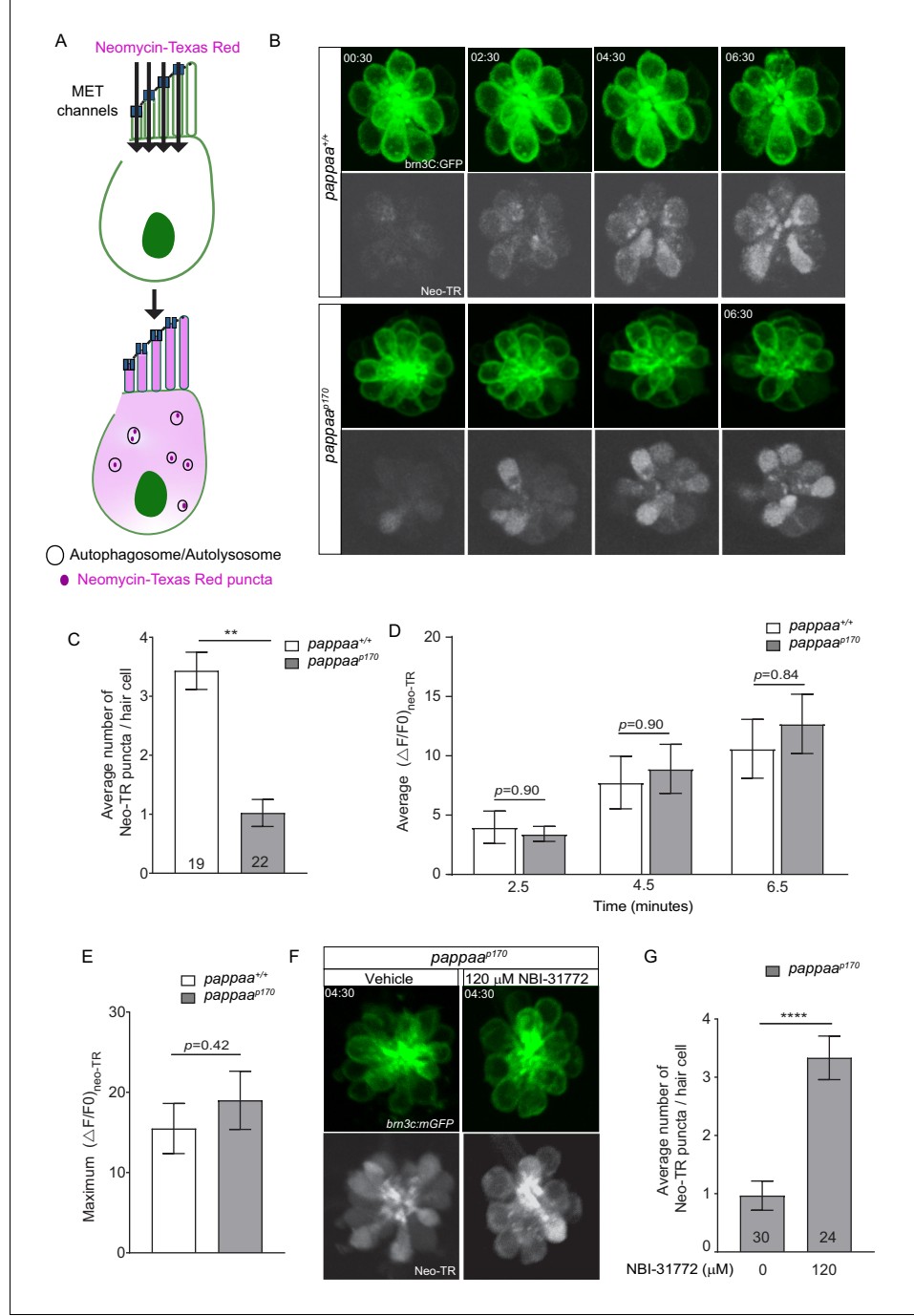

**Figure 4.** Pappaa regulates neomycin-induced autophagy. (**A**) Schematic showing cell entry and autophagy of neomycin-Texas Red. (**B**) Representative time lapse images of *brn3C:mGFP*-labeled neuromast hair cells (green) at 5 dpf following exposure to 10 µM neomycin-Texas Red (white). (**C**) Mean number of neomycin-Texas Red puncta/hair cell in wild-type and *pappaa^p170^* larvae at 5 dpf. **p<0.01 t-test, Mann–Whitney correction. N = 19 hair cells (wild type) and 22 hair cells (*pappaa^p170^*) collected from four larvae/genotype. (**D**) Mean neomycin-Texas Red ΔF/F0 at 2.5, 4.5, and 6.5 min post-exposure. Multiple *t*-test with Holm–Sidak correction found no significant difference. N = 22 hair cells (wild type) and 22 hair cells (*pappaa^p170^*) collected from four larvae/genotype. (**E**) Maximum change in neomycin-Texas Red fluorescent intensity across treatment time. Unpaired t-test with Mann–Whitney correction found no significant difference. N = 22 hair cells (wild type) and 22 hair cells (*pappaa^p170^*) collected from four larvae/genotype. See *Videos 1* and *2*. (**F**) Representative time lapse images of vehicle or 120 µM NBI-31772 treated *brn3C:mGFP*-labeled neuromast hair cells (green) of *pappaa^p170^* larvae at 5 dpf following exposure to 10 µM neomycin-Texas Red (white). (**G**) Mean number of neomycin-Texas Red puncta/hair cell in

*Figure 4 continued*

vehicle or 120 µM NBI-31772 treated *pappaa*^*p170* larvae at 5 dpf. ****p<0.0001 t-test, Mann–Whitney correction. N = 30 hair cells (Vehicle) and 24 hair cells (120 µM NBI-31772) collected from four larvae/ group. Error bars=SEM. The online version of this article includes the following source data for figure 4:

**Source data 1.** Mean number of neomycin-Texas Red puncta in wild-type and *pappaa*^*p170* lateral line hair cells.
**Source data 2.** Mean change in neomycin-Texas Red fluorescence over time in wild-type and *pappaa*^*p170* lateral line hair cells.
**Source data 3.** Maximum change in neomycin-Texas Red fluorescence in wild-type and *pappaa*^*p170* lateral line hair cells.
**Source data 4.** Mean number of neomycin-Texas Red puncta in NBI-31772-treated *pappaa*^*p170* lateral line hair cells.

'free' IGF1 levels were reduced (*Wilczak et al., 2003*). As an upstream regulator of IGF1 signaling that is spatially restricted, Pappaa may provide a more viable therapeutic alternative.

# Materials and methods

## Key resources table

| Reagent type (species) or resource | Designation | Source or reference | Identifiers | Additional information |
|---|---|---|---|---|
| Gene (*Danio rerio*) | *pappaa*^*p170* | PMID:25754827 | RRID: ZFIN_ZDB-GENO-190322-4 | Single-nucleotide nonsense mutation C>T at position 964 in Exon 3 |
| Strain, strain background (*Danio rerio*) | *Tg(brn3c:GFP)* | PMID:15930106 | RRID: ZFIN_ZDB-ALT-050728-2 | |
| Strain, strain background (*Danio rerio*) | *TLF* | Zebrafish International Resource Center (ZIRC) | RRID: ZFIN_ZDB-GENO-990623-2 | |
| Antibody | anti-KDEL (mouse monoclonal) | Calbiochem | AB_212090 | 1:500 |
| Antibody | anti-GFP (rabbit polyclonal) | ThermoFisher Scientific | RRID:AB_221569 | 1:500 |
| Antibody | Alexa 488, secondary (rabbit polyclonal) | ThermoFisher Scientific | RRID:AB_2576217 | 1:500 |
| Other | mitotracker greenFM | ThermoFisher Scientific | Catalog number: M7514 PubChem CID:70691021 | 100 nM for 5 min |
| Other | 0.25% trypsin-EDTA | Sigma-Aldrich | Catalog number: T3924 | |
| Other | TRIzol | Invitrogen | Catalog number: 15596026 | |
| Other | Texas Red | ThermoFisher Scientific | Catalog number: T6134 | |
| Other | Sso fast Eva Green Supermix | Bio-Rad | Catalog number: 1725200 | |
| Chemical compound, drug | Neomycin | Sigma-Aldrich | Catalog number: N1142 | |
| Chemical compound, drug | NBI-31772 | Fisher Scientific | Catalog number: 519210 | |
| Chemical compound, drug | Thapsigargin | Tocris | Catalog number: 1138 | |
| Chemical compound, drug | Tunicamysin | Caymen Chemical | Catalog number: 11445 | |
| Commercial assay or kit | SuperScript II Reverse Transcriptase | Invitrogen | Catalog number: 18064014 | |
| Software, algorithm | Fluoview (FV10-ASW 4.2) | Olympus | RRID:SCR_014215 | |
| Software, algorithm | Imaris | Bitplane | RRID:SCR_007370 | |
| Software, algorithm | ImageJ | PMID:22743772 | RRID:SCR_003070 | |
| Software, algorithm | GraphPad PRISM | graphpad | RRID:SCR_002798 | |
| Software, algorithm | JMP Pro 15.0 | SAS Institute Inc | RRID:SCR_014242 | |

## Animals

To generate *pappaa*$^{+/+}$ and *pappaa*$^{p170}$ larvae, adult *pappaa*$^{p170/+}$ zebrafish (on a mixed Tubingen long-fin [*TLF*], *WIK* background) were crossed into *Tg(brn3c:mGFP)*$^{s356t}$ and then incrossed. Embryonic and larval zebrafish were raised in E3 media (5 mM NaCl, 0.17 mM KCl, 0.33 mM CaCl$_2$, 0.33 mM MgSO$_4$, pH adjusted to 6.8–6.9 with NaHCO$_3$) at 29°C on a 14 hr/10 hr light/dark cycle through 5 dpf (*Gyda et al., 2012*; *Kimmel et al., 1995*). *pappaa*$^{p170}$ larvae were obtained from crosses of *pappaa*$^{p170/+}$ adults. Mutant larvae were identified either by lack of swim bladder inflation (*Wolman et al., 2015*) or by genotyping after the experiment. For genotyping, PCR was performed as previously described (*Wolman et al., 2015*) using forward primer: AGACAGGGATGTGGAG TACG, and reverse primer: GTTGCAGACGACAGTACAGC. PCR conditions were as follows: 3 min at 94°C, followed by 40 cycles of 94°C for 30 s, 57°C for 1 min, and 70°C for 1 min. The PCR product was digested with MseI (New England Biolabs R0525S) resulting in a 245 bp mutant restriction fragment that is distinct from the 271 bp wild-type restriction fragment. The restriction digest reaction was run on a 3% agarose gel.

## Pharmacology

All pharmacological experiments were performed on *Tg(brn3c:mGFP)* larvae. Compounds were added to the larval E3 media at 4–5 dpf. Thapsigargin (Tocris No. 1138; dissolved in DMSO) was added at 10–20 µM for 1 hr at 5 dpf or at 0.3–1 µM for 24 hr at 4 dpf. Tunicamycin (Caymen Chemical 11445; dissolved in DMSO) was added at 2–3 µM for 24 hr, beginning at 4 dpf. To stimulate IGF1 bioavailability, larvae were pre-treated with 100–120 µM NBI-31772 (Fisher Scientific 519210; dissolved in DMSO) for 24 hr starting from day 4, then exposed to 20 µM thapsigargin for 1 hr. Optimal drug concentrations and exposure times were determined from pilot experiments with a wide range of doses. The highest dose that yielded hair cell death but was not lethal to the larvae was chosen as the upper limit for the dose ranges used in experiments. Following each treatment period, larvae were washed three times with E3 before fixation with 4% paraformaldehyde (diluted to 4% w/v in phosphate buffered saline (PBS) from 16% w/v in 0.1 M phosphate buffer, pH 7.4). Vehicle-treated controls were treated with 0.1% DMSO in E3.

## Induction of a dominant negative IGF1R

To attenuate IGF1R signaling, we used *Tg(hsp70:dnIGF1Ra-GFP)* larvae that express *dnIGF1Ra-GFP* under the control of zebrafish hsp70 promoter (*Kamei et al., 2011*). Expression of *dnIGF1Ra-GFP* was induced by a 1 hr heat shock at 37°C at 4.5 dpf followed by another heat shock 12 hr later at 5 dpf. To control for any possible heat shock effects, non-transgenic controls were subjected to the same treatment. Thapsigargin was added 3 hr after the second heat shock at 5 dpf.

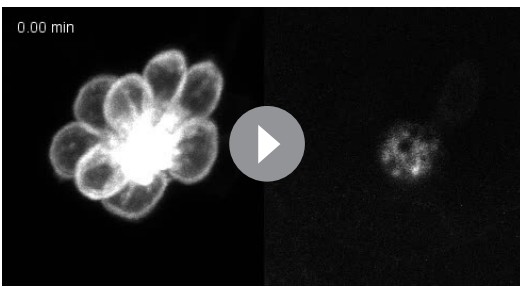

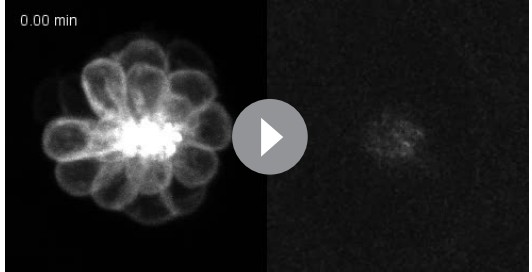

**Video 3.** Wild-type lateral line hair cells following exposure to neomycin-Texas red. This video was constructed from time lapse images of wild-type *brn3c: mGFP* lateral line hair cells (left) following exposure to 10 µM neomycin-Texas red (right).
https://elifesciences.org/articles/59687#video3

**Video 4.** *pappaa*$^{p170}$ lateral line hair cells following exposure to neomycin-Texas red. This video was constructed from time lapse images of *pappaa*$^{p170}$ *brn3c:mGFP* lateral line hair cells (left) following exposure to 10 µM neomycin-Texas red (right).
https://elifesciences.org/articles/59687#video4

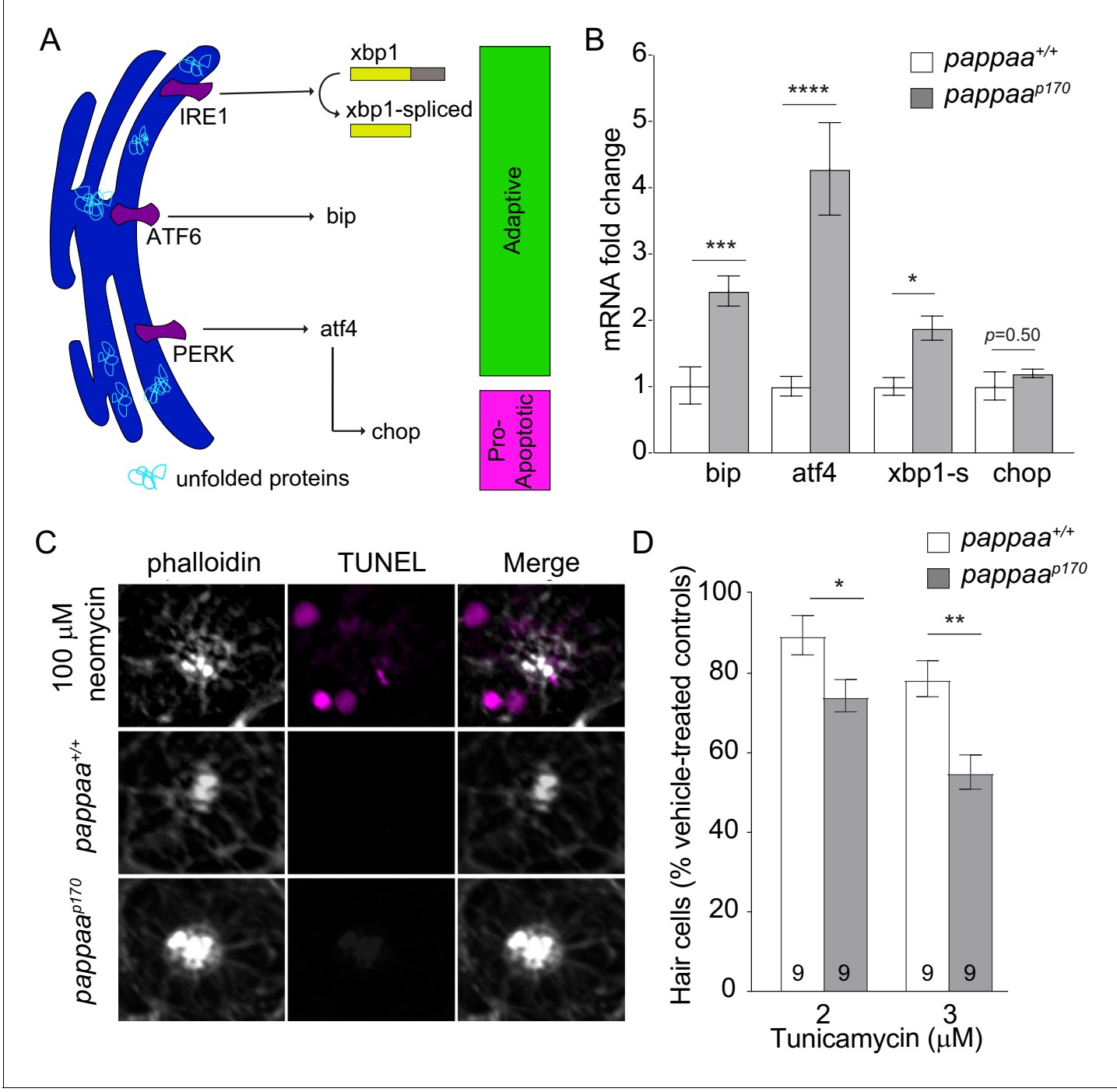

**Figure 5.** Pappaa loss causes ER stress. (**A**) Schematic of the UPR pathway. The accumulation of unfolded proteins activates the UPR receptors, IRE1, ATF6, and PERK, signifying ER stress. In the early adaptive phase of ER stress, the UPR promotes cell survival through the upregulation of pro-survival factors including *bip*, *atf4*, and *spliced xbp1*. A switch from an adaptive to a pro-apoptotic UPR occurs during the late phase of ER stress in which Chop, a pro-apoptotic transcription factor, is upregulated. (**B**) Mean fold change in UPR mRNA levels in wild-type and *pappaa^p170* hair cells at 5 dpf. N = 2–3 technical replicates/gene. *p<0.05, ***p<0.001, ****p<0.0001, two-way ANOVA, Holm–Sidak post-test. Error bars=SEM. (**C**) Representative images of TUNEL staining (magenta) in wild-type and *pappaa^p170* lateral line hair cells. Stereocilia are counterstained with phalloidin (white). A 30 min treatment with 100 µM neomycin was used as positive control (top). (**D**) Mean percentage of surviving hair cells following a 24 hr treatment with tunicamycin starting from 4 dpf. To calculate hair cell survival percentage, hair cell number post-treatment was normalized to mean hair cell number in vehicle-treated larvae of the same genotype. *p<0.05, **p<0.01, two-way ANOVA, Holm–Sidak post-test. N = 8–10 larvae per group (shown at base of bars), three neuromasts/larva from two experiments were analyzed. Total number of neuromasts included in the analysis = 51 (wild type;

*Figure 5 continued*

vehicle treated), 57 (*pappaa*[*p170*]; vehicle treated), 27 (wild type; 2 µM Tunicamycin), 27 (*pappaa*[*p170*]; 2 µM Tunicamycin), 27 (wild type; 3 µM Tunicamycin), 27 (*pappaa*[*p170*]; 3 µM Tunicamycin). Error bars=SEM.

The online version of this article includes the following source data for figure 5:

**Source data 1.** Mean fold change in UPR transcript levels in wild-type and *pappaa*[*p170*] lateral line hair cells.
**Source data 2.** Hair cell survival following treatment with Tunicamycin in wild-type and *pappaa*[*p170*] larvae.

## Immunohistochemistry

To visualize hair cells, 5 dpf larvae were fixed in 4% paraformaldehyde for 1 hr at room temperature and then rinsed three times with PBS. Larvae were blocked for 1 hr at room temperature in incubation buffer (IB: 0.2% bovine serum albumin, 2% normal goat serum, 0.8% Triton-X, 1% DMSO, in PBS, pH 7.4). Primary antibodies included anti-GFP to visualize *brn3c:mGFP* hair cells (1:500, rabbit polyclonal; ThermoFisher Scientific, RRID:AB_221569), phosphorylated IGF1R anti-IGF1 receptor phospho Y1161, 1:100, rabbit IgG; Abcam; (*Chablais and Jazwinska, 2010*), and anti-KDEL to visualize the ER (1:500, mouse monoclonal; Calbiochem, RRID: AB_212090). Larvae were incubated in primary antibodies in IB overnight at 4°C. Larvae were then rinsed five times for 10 min with IB. Following that, larvae were incubated in fluorescently conjugated secondary antibodies in IB for 4 hr at room temperature and rinsed five times with IB. Secondary antibodies included AlexaFluor488-conjugated antibody (goat anti-rabbit polyclonal, 1:500; ThermoFisher Scientific, RRID:AB_2576217) and AlexaFluor594-conjugated secondary antibody (goat anti-rabbit polyclonal, 1:500; ThermoFisher Scientific RRID:AB_142057). Larvae were mounted in 70% glycerol in PBS. Images were acquired with an Olympus Fluoview confocal laser scanning microscope (FV1000) using Fluoview software (FV10-ASW 4.2).

Quantification of the percentage area occupied by KDEL immunofluorescence was done using ImageJ. Maximum-intensity projections of z-stacks that spanned the entire depth of the neuromast were generated. A binary mask for the KDEL channel was generated using automated Otsu thresholding values, and the area fraction was measured. pIGF1R fluorescence was measured using ImageJ (*Schneider et al., 2012*) by drawing a region of interest around brn3c:GFP-labaled hair cells or 10 randomly selected SOX2-labeled support cells of the neuromast from Z-stack summation projections that included the full depth of the neuromast. Background fluorescence intensity was measured by drawing a region of interest away from the neuromast in the same Z-stack summation projection. The corrected total cell fluorescence (CTCF) was used to subtract background fluorescence. The CTCF formula was as follows: Integrated Density − (Area of selected cells × Mean fluorescence of background) (*McCloy et al., 2014*).

## Hair cell survival

Hair cell survival experiments were performed in *Tg(brn3c:mGFP)* and *Tg(hsp70:dnIGF1Ra-GFP)* larvae as previously described (*Alassaf et al., 2019*). Hair cells from three stereotypically positioned neuromasts (IO3, M2, and OP1) (*Raible and Kruse, 2000*) were counted from z-stacks and averaged for each larva. Hair cell survival was calculated as a percentage with the following formula: [(mean number of hair cells after treatment)/(mean number of hair cells in vehicle treated group)] × 100.

## RT-qPCR

Fluorescent sorting of *brn3c:mGFP* hair cells, RNA extraction, cDNA synthesis, and RT-qPCR were performed as previously described (*Alassaf et al., 2019*). For each genotype, 200 five dpf *Tg(brn3c: GFP)* larvae were incubated for 15 min in Ringer's solution (116 mM NaCl, 2.9 mM KCl, 1.8 mM CaCl$_2$, 5 mM HEPES, pH 7.2) (*Guille, 1999*) prior to dissection. *pappaa*[*p170*] larvae were identified by lack of swim bladder inflation (*Wolman et al., 2015*). Tails were dissected and transferred into 1.5 mL tubes containing Ringer's solution on ice. For cell dissociation, dissected tails were incubated in 1.3 mL of 0.25% trypsin–EDTA (Sigma-Aldrich) for 20 min and triturated gently by P1000 pipette tip every 5 min. To stop cell digestion, 200 µL of stop solution containing 30% fetal bovine serum and 6 mM CaCl$_2$ in PBS solution was added to the samples (*Steiner et al., 2014*). The samples were centrifuged at 400 g for 5 min at 4°C, and the supernatant was removed. Ca$^{2+}$-free Ringer's solution (116

mM NaCl, 2.9 mM KCl, 5 mM HEPES, pH 7.2) was added to rinse the cell pellet, and the samples were centrifuged again. Finally, the cell pellet was resuspended in $Ca^{2+}$-free Ringer's solution and kept on ice until ready to FAC sort. Cells were filtered through a 40 µm cell strainer. A two-gate sorting strategy was used. Cell vitality was assessed by DAPI, followed by forward scatter and GFP gate to isolate GFP+ cells. Live and GFP+ cells were collected into RNAse-free tubes containing 500 µL of TRIzol reagent (Invitrogen) for total RNA extraction. cDNA was synthesized using SuperScript II Reverse Transcriptase (Invitrogen 18064014). Real-time quantitative PCR (RT-qPCR) was performed using Sso fast Eva Green Supermix (Bio-Rad 1725200) in a StepOnePlus Real-Time PCR System (Applied Biosystems) based on manufacture recommendation. Reactions were run in three to four technical replicates containing cDNA from 50 ng of total RNA/reaction. The primer sequences for the UPR genes are as follows: for bip, forward: ATCAGATCTGGCCAAAATGC and reverse: CCACG TATGACGGAGTGATG; atf4, forward: TTAGCGATTGCTCCGATAGC and reverse: GCTGCGG TTTTATTCTGCTC; for chop, forward: ATATACTGGGCTCCGACACG and reverse: GATGAGGTG TTCTCCGTGGT (*Howarth et al., 2014*); for xbp1-spliced, forward: TGTTGCGAGACAAGACGA and reverse: CCTGCACCTGCTGCGGACT (*Vacaru et al., 2014*); for b-actin (endogenous control), forward TACAGCTTCACCACCACAGC and reverse: AAGGAAGGCTGGAAGAGAGC (*Wang et al., 2005*). Cycling conditions were as follows: 1 min at 95℃, then 40 cycles of 15 s at 95℃, followed by 1 min at 60℃ (*Jin et al., 2010*). Analysis of relative gene expression was done using the $2^{-\Delta\Delta Ct}$ method (*Livak and Schmittgen, 2001*).

## MitoTracker

To assess mitochondrial morphology, 5 dpf larvae were incubated in 100 nM mitotracker green FM (ThermoFisher Scientific M7514; dissolved in anhydrous DMSO) for 5 min. Following the incubation period, larvae were washed three times in E3, anesthetized in 0.02% tricaine (Sigma-Aldrich) in E3, and mounted as previously described (*Stawicki et al., 2014*). Larvae were transferred into a Nunc Lab-Tek chambered glass (ThermoFisher scientific 155379PK) and stabilized under a nylon mesh and two slice anchors (Warner instruments). To avoid inconsistencies in mitotracker retention over time, each larva was individually stained and then immediately imaged. Z-stack images were acquired using an Olympus Fluoview confocal laser scanning microscope (FV1000) with a 60× oil-immersion objective lens and Fluoview software (FV10-ASW 4.2). A maximum-intensity projection of Z-stacks that covered the full depth of the neuromast hair cells was used for ImageJ analysis. For each neuromast, the maximum-intensity projection was first inverted, sharpened, and then automatically thresholded before the 'analyze particles' function was used to measure the average mitochondrial circularity. Orthogonal view and 3D rendering were obtained using Imaris software (Bitplane).

## Neomycin-Texas Red

To examine autophagy, Conjugation of Texas Red-X-succinimidyl ester to neomycin sulfate hydrate (Sigma-Aldrich) was done following the previously described protocol (*Stawicki et al., 2014*). To make a final concentration of 10 µM of neomycin-Texas Red, neomycin sulfate hydrate was dissolved in nuclease-free water to 50% of the final solution volume. 0.5 M $K_2CO_3$ at pH 9.0 was added at 17.6% final volume. Texas Red-X-succinimidyl ester (Life Technologies) was dissolved in dimethylformamide at 2.5 mM and was added at 12% final volume. The volume of the mixture was brought to 100% with deionized water and the solution incubated overnight at 4℃ to allow the conjugation reaction to go to completion. Larvae were anesthetized in 0.02% tricaine (Sigma-Aldrich) in E3 and transferred to the imaging chamber. Z-stack images of each neuromast were acquired at baseline (prior to Neo-TR exposure) for 2.5 min at 30 s intervals. Following the addition of neomycin-Texas Red to the larva, images were acquired at 30 s intervals for 1 hr. Hair cells within each neuromast that were visually accessible with no significant physical overlap with neighboring hair cells were selected for analysis. Neomycin-Texas Red puncta were counted manually for each hair cell. For the rescue experiment, larvae were treated with 120 µM NBI-31772 for 24 hr starting from 4 dpf. At 5 dpf, larvae were anesthetized in 0.02% tricaine in 133 µM NBI-31772 and transferred to the imaging chamber. Neomycin-Texas Red was added to a final concentration of 120 µM NBI-31772. Z-stack summation projections that included the full depth of the neuromast hair cells were used to quantify fluorescent intensity of neomycin-Texas Red. A region of interest around *brn3c:GFP*-labaled hair cells was drawn with the free hand tool in ImageJ. Background fluorescent intensity was measured

by drawing a ROI away from the neuromast in the same Z-stack summation projection. The CTCF was used to subtract background fluorescence. The CTCF formula was as follows: Integrated Density – (Area of selected cells × Mean fluorescence of background) (*McCloy et al., 2014*). F0 is the CTCF value at baseline (before the addition of Neomycin-Texas Red), and F is the CTCF value after the addition of Neomycin-Texas Red. The ΔF/F0 was calculated for each time point as follows: the CTCF value after the addition of Neomycin-Texas Red (F)/the CTCF value at baseline (F0). Time lapse movies were made using ImageJ and corrected for xy drift using the ImageJ plugins StackReg and TurboReg (*Thévenaz et al., 1998*).

## Electron microscopy

Five dpf *pappaa*$^{+/+}$ and *pappaa*$^{p170}$ larvae in a Tubingen long-fin (*TLF*) background were processed as previously described (*Allwardt et al., 2001*). Larvae were fixed for 15 min at 4°C using 1% paraformaldehyde, 1.6% glutaraldehyde, 0.15 mm $CaCl_2$, and 3% sucrose in 0.06 M phosphate buffer, pH 7.4. Larvae were then rinsed and post-fixed in osmium tetroxide (2% in phosphate buffer) for 30 min at 4°C then 1.5 hr at room temperature. The larvae were rinsed in phosphate buffer and in maleate buffer (0.05 M, pH 5.9) and then processed in a solution of 2% uranyl acetate in maleate buffer. Larvae were dehydrated in a graded series of ethanol concentrations. Larvae were then incubated in propylene oxide for 20 min, followed by infiltration with Araldite/Epon resin. Lastly, larvae were cured for 72 hr at 60°C. Eighty nanometer thick sections were mounted on slot grids and post-stained with lead citrate and saturated uranyl acetate. Images were acquired with a Philips CM120 scanning transmission electron microscope with a BioSprint 12 series digital camera using AMT Image Capture Software Engine V700 (Advanced Microscopy Techniques). All images were of head neuromasts, particularly IO 1–3 and SO 1–3. Mitochondrial morphology and ER–mitochondria associations were analyzed using ImageJ (*Schneider et al., 2012*). For mitochondrial morphology, a ROI was manually drawn around each mitochondrion using the free hand tool. The 'Analyze Particles' function was used to measure the area, perimeter, and circularity. Mitochondrial interconnectivity was calculated as the area/perimeter (*Wiemerslage and Lee, 2016*). To measure the inter-organelle distance, the line segment tool was used to draw a straight line connecting the two closest points between the ER and mitochondria. Any ER–mitochondria associations with distances exceeding 100 nm were discarded from the final analysis. The frequency of ER–mitochondria associations was determined by manually counting the number of ER fragments that were within 100 nm of each mitochondrion.

## Statistics

All data were analyzed using GraphPad Prism Software 7.0b for statistical significance (GraphPad Software Incorporated, La Jolla, CA, RRID:SCR_002798). Data are presented as the mean ± standard error of the mean (SEM). The statistical tests and sample size for each experiment are stated in the figure legends. Power analysis of sample size using JMP Pro 15.0 software (SAS Institute Inc) revealed that all our experiments had sufficient power to avoid types 1 and 2 errors. The assumption of normality was tested using Shapiro-Wilk's test. Parametric analyses were performed using a two-tailed unpaired t-test with Welch's correction, multiple t-tests with a Holm–Sidak correction, or two-way ANOVA with a Holm–Sidak correction. Non-parametric analyses were performed using a two-tailed t-test with Mann–Whitney correction. Significance was set at $p < 0.05$. All data presented are from individual experiments expect for data in *Figures 1F*, *2D,* and *5C*. Hair cell survival data collected from multiple experiments were normalized to their respective controls.

## Acknowledgements

The authors would like to thank Benjamin August (UW School of Medicine and Public Health electron microscope facility) for his technical expertise, Dr. Yevgenya Grinblat (University of Wisconsin-Department of Integrative Biology) for the use of the RT-qPCR cycler, Emily Daykin for her assistance with data acquisition, and Daniel North for fish care and maintenance.

## Additional information

### Funding

| Funder | Grant reference number | Author |
|---|---|---|
| Ministry of Education – Kingdom of Saudi Arabi | Graduate Student Fellowship | Mroj Alassaf |
| University of Wisconsin-Madison | UW internal funds | Mary C Halloran |

The funders had no role in study design, data collection and interpretation, or the decision to submit the work for publication.

### Author contributions

Mroj Alassaf, Conceptualization, Data curation, Formal analysis, Investigation, Writing - original draft; Mary C Halloran, Resources, Supervision, Funding acquisition, Project administration, Writing - review and editing

### Author ORCIDs

Mroj Alassaf (iD) https://orcid.org/0000-0001-6277-9417
Mary C Halloran (iD) https://orcid.org/0000-0001-6086-5928

### Ethics

Animal experimentation: This study was performed in accordance with the recommendations in the Guide for the Care and Use of Laboratory Animals of the National Institutes of Health. Animals were handled according to approved institutional animal care and use committee (IACUC) protocols (L005704) of the University of Wisconsin.

### Decision letter and Author response

Decision letter https://doi.org/10.7554/eLife.59687.sa1
Author response https://doi.org/10.7554/eLife.59687.sa2

## Additional files

### Supplementary files

• Transparent reporting form

### Data availability

All data generated or analyzed during this study are included in the manuscript and supporting files. Source data files are provided for Figures 1–5, and Figure 2—figure supplement 1.

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
