## [Decision Letter]

Thank you for submitting your article "Pregnancy associated plasma protein-aa regulates endoplasmic reticulum-mitochondria associations" for consideration by *eLife*. Your article has been reviewed by 3 peer reviewers, and the evaluation has been overseen by a Reviewing Editor and Didier Stainier as the Senior Editor. The following individuals involved in review of your submission have agreed to reveal their identity: Alison Coffin (Reviewer #2); Katie Kindt (Reviewer #3).

The reviewers have discussed the reviews with one another and the Reviewing Editor has drafted this decision to help you prepare a revised submission.

Summary:

This current study builds on previous work from the same group published in e*Life*. This past work focused on the mechanism that renders lateral line hair cells of pappaa mutants more susceptible to the ototoxin neomycin. This work found that mitochondrial dysfunction was the underlying cause for neomycin susceptibility. This current study expands on the previous work and suggests that not only defects in mitochondria, but also the ER are involved in neomycin susceptibility. The authors use a variety of approaches including TEM, live imaging, pharmacology and RT-qPCR in their present study. Using TEM the authors show that mitochondria – ER associated are more numerous. Furthermore, similar to disrupting mitochondrial calcium pharmacologically, disrupting ER calcium also renders Pappaa-deficient hair cells more susceptible to neomycin. The authors suggest that this ER dysfunction manifests in several ways. They use live imaging to show that in pappaa mutants hair cells are unable to properly package neomycin into autosomes. In addition, via RT-qPCR they show that pappaa mutants have an increased unfolded protein response (UPR). Currently the relationship between all of these pathological issues is unclear, but this work does reveal additional mechanisms that could render loss of Pappaa detrimental to hair cell health. Although the work is well written and presented and statistically sound, it there are several experiments that are needed to strengthen the claims presented in this study.

Essential revisions:

1. Location of TEM micrographs in hair cell

The morphology of organelles can vary based on location within the cell. For example, in hair cells the ER near the nucleus can be distinct from the ER present near the contacts made with efferent neurons or afferent neurons. (https://pubmed.ncbi.nlm.nih.gov/1430341/; https://physoc.onlinelibrary.wiley.com/doi/10.1113/jphysiol.2013.267914).

Can the authors indicate what direction the sections (apical-basal or transverse) were taken, where in the hair cells are the sections were taken and how they determined this location?

2. Quantification of mitochondrial fragmentation

It is clear from the TEM cross sections that the mitochondria in hair cells (Figure 3 A) are quite different between pappaa mutants and controls. Whether there are mitochondria or ER networks are present is not apparent from these TEM images. Nor is it entirely clear that the networks are fragmented. The authors use plugins developed for confocal imaging to estimate fragmentation base on circularity and area/perimeter measurement. It is unclear if these measurement translate to hair cells or TEM. In addition to fragmentation in TEM images, the fragmented mitochondria in pappaa mutants are also hard to see in the live, max-projected mitoTimer images.

The mitochondrial networks and fragmentation may be clearer or be better quantified by acquiring super resolution images of hair cells labeled with mitoTracker. In addition, it is possible that the fragmentation may also be visible or more convincing in videos of Z-stacks of mitoTracker label compared to in the max-projected images provided.

3. Examination of hair cell ER morphology

The previous work on Pappaa in zebrafish hair cells focused extensively on the mitochondria while the currently study the shifted the focus to the hair cell ER. While the ER-mito distances are convincing, a more wholistic picture of the amount or distribution of the ER in wildtype and mutant is lacking.

This could be accomplished either using a transgenic line that labels the ER or a KDEL antibody (https://www.ncbi.nlm.nih.gov/pmc/articles/PMC4007406/).

4. It qualitatively appears that pappaa mutant hair cells are taking up a greater quantity of fluorescent Neo faster than WT i.e. the fluorescent intensity is greater in more hair cells. Did the authors quantify Neo-TR uptake?

5. Specificity of the pharmacological treatments

The authors perform numerous pharmacological experiments to disturb ER calcium. The authors suggest that their pharmacological manipulations trigger hair cell death due to the alteration in the interplay between ER/mito calcium in hair cells. What concentration of either of these drugs does it take to kill WT hair cells? Dose-response curves comparing WT and mutant would help support the idea that hair-cell death observed is a direct effect of the drugs on hair-cell ER-mitochondria calcium signaling.

Pharmacology is non-cell autonomous and the authors do not present evidence that these compounds specifically impact hair cell ER or mitochondrial calcium. Alternatively, these compound could impact supporting cell ER (https://elifesciences.org/articles/52160) as well as the ER in the innervating afferent or efferent neurons.

More direct evidence show that hair cell mitochondria or ER calcium (measurements using mitoGCaMP such as in the previous study) are impacted by these treatments would make the author's claims more compelling.

6. The disconnect between IGFR1 and results in the current study.

The identify and location of IGFR1 and the IGFBP are still undefined in this system and therefore it remains unclear exactly how IGRR1 or Pappaa impact sensory hair cells. In previous work on pappaa mutants (enhanced startle response, defects in photoreceptor synapse formation, defects in hair cell mitochondria) the role of IGFR1 in these processes was validated. In the current study, the link with IGFR1 is implied throughout.

It is true that the relationship between IGFR1 and Pappa is well characterized and that currently the only known substrates of Pappa are IGFBPs. Despite this work, it is still possible that given the range of phenotypes in pappaa mutants, that Pappa has other protein substrates that have not yet been identified, or other has biological functions unrelated to the IGF system.

To verify IGFR1 in this current study the authors could use NB1-31772 to stimulate IGF1 bioavailability and test whether this rescues either the autophagy or UPR defects in pappaa mutants. Being able to rescue these phenotypes also makes the study more compelling.

7. The authors state that there is not more spontaneous hair cell death in pappaa mutants compared to controls (line 443). Previous work has shown in zebrafish that Usher mutants (cdh23, ush1c, myo7a) also have an early UPR (https://www.ncbi.nlm.nih.gov/pmc/articles/PMC4007406/). Similar to pappaa mutants usher mutants have the same # of hair cells compared to controls, indicating no spontaneous hair cell death. But interestingly Usher mutants do have more TUNEL positive hair cells compared to controls, indicating that more hair cells in Usher mutants are in the process of apoptosis. Based on this new finding implicating the UPR response in pappaa mutants, could pappaa mutants, similar to hair cells in Usher mutants be more fragile (neomycin susceptible) as they are more likely to be in the process of apoptosis? A TUNEL label in pappaa mutants could reveal this. In addition, this paper on UPR in Usher mutant hair cells could be a useful paper to add to the discussion.

8. Line 445-451: "Together, these findings suggest that Pappaa may regulate ER-mitochondria associations by promoting ER homeostasis. It is important to note that the ER and mitochondria are engaged in a constant feedback loop." This line of reasoning seems rather circular, considering that the previous study showed Pappaa regulates mitochondrial function. If mitochondrial function is impaired, it seems likely that ER homeostasis would be disrupted as well.

9. Methods: Overall, the methods section needs more detail. All experiments that were not previously performed by the author or the author's lab should have a concise description of what the authors did next to the reference (e.g. fish were imaged under Lab-Tek Chambered Coverglass (Fisher Scientific) where they were immobilized under a nylon mesh and two stainless-steel slice hold-downs (Warner Instruments) per Stawicki et al., 2014) A detailed description of how individual pappaa170 larvae used in experiments were genotyped is needed. A comprehensive description of how mitochondrial circularity was measured using the "mitochondrial morphology" plug-in in ImageJ is needed.

10. Statistics: how did the authors determine the power of the experiments were sufficient to avoid Type I and Type II error?

[Editors' note: further revisions were suggested prior to acceptance, as described below.]

Thank you for resubmitting your work entitled "Pregnancy associated plasma protein-aa regulates endoplasmic reticulum-mitochondria associations" for further consideration by *eLife*. Your revised article has been reviewed by 2 peer reviewers and the evaluation has been overseen by Didier Stainier as the Senior Editor, and a Reviewing Editor.

The manuscript has been improved but there are some remaining issues that need to be addressed before acceptance, as outlined below:

This current study builds on previous work from the same group published in *eLife*. This past work focused on the mechanism that renders lateral line hair cells of pappaa mutants more susceptible to the ototoxin neomycin. This previous study found that mitochondrial dysfunction was the underlying cause for neomycin susceptibility. This current study expands on the previous work and suggests that not only defects in mitochondria, but also the ER are involved in neomycin susceptibility. The authors use a variety of approaches including TEM, live imaging, pharmacology and RT-qPCR in their present study. Using TEM the authors show that mitochondria – ER associations are more numerous. Furthermore, similar to disrupting mitochondrial calcium pharmacologically, disrupting ER calcium also renders Pappaa-deficient hair cells more susceptible to neomycin. The authors suggest that this ER dysfunction manifests in several ways. They use live imaging to show that in pappaa mutants hair cells are unable to properly package neomycin into autosomes. In addition, via RT-qPCR they show that pappaa mutants have an increased unfolded protein response (UPR). The work is well written and presented and statistically sound.

Overall, this revision addressed a number of specific concerns experimentally, including providing pharmacological evidence that IGF system is involved in the hair cell processes they're examining. However some issues still need to be addressed:

1. More is needed is needed on the locus of action where Pappaa/ IGF signaling regulates the hair cell ER-mitochondria axis. The first author has previously reported pappaa expression to be in supporting cells. However igf1rb expression in neuromasts has also been reported to be predominantly in supporting cells (e.g. https://piotrowskilab.shinyapps.io/neuromast_homeostasis_scrnaseq_2018/). If Pappa/IGF signaling is occurring in supporting cells, the effect on hair cells would be indirect. Resolution of where signaling occurs would be a valuable addition to this study. At the very least, this issue warrants discussion in the manuscript.

2. Currently the relationship between the numerous pathological issues present in pappaa hair cells (mitochondria and ER morphology disruption, lysosomal packaging) is unclear. Does the origin of the deficit rest in the mitochondria, the ER or another organelle? Overall, this work does reveal an additional mechanism (ER calcium) that renders loss of Pappaa detrimental to hair cell health. One caveat of the study is that it relies heavily on pharmacology which impacts not only the ER in hair cells, but also the glial-like supporting cells that surround hair cells, as well as the afferents and efferents that innervate this sensory system.

These points would be worth discussion.

3. The authors indicated in Figure 1A the orientation of the TEM sections and identified which anterior neuromasts. But it was not entirely clear how they determined the location of their organelles relative to neuronal contacts. There was no discussion in the manuscript on how they analyzed the images using the post-synaptic density and synaptic ribbons as landmarks. Additionally, there was no mention of efferent contacts in either the manuscript or the rebuttal.

4. KDEL labeling does not appear to be specifically labeling ER in the images shown in Figure 1F. Instead, the labeling appears completely colocalized with hair cell plasma membrane bound GFP. For comparison, Blanco-Sánchez et al., 2014 showed discrete labeling of membrane within hair cells. Specificity of labeling should be verified if KDEL labeling will be used to measure the amount or distribution of the ER in wildtype and pappaa mutants.

5. The authors did not answer the question of how they determined their experiments were sufficiently powered.

6. The x/z and y/z max intensity images of Mitotracker are better illustrations of what appear to be more fragmented mitochondria. The Imaris 3D reconstructions the authors mentioned in the rebuttal would have been informative, but they did not include them in Figure 3.

7. The P values for non-significance should be reported.

---

## [Author Response]

Essential revisions:1. Location of TEM micrographs in hair cellThe morphology of organelles can vary based on location within the cell. For example, in hair cells the ER near the nucleus can be distinct from the ER present near the contacts made with efferent neurons or afferent neurons. (https://pubmed.ncbi.nlm.nih.gov/1430341/; https://physoc.onlinelibrary.wiley.com/doi/10.1113/jphysiol.2013.267914).Can the authors indicate what direction the sections (apical-basal or transverse) were taken, where in the hair cells are the sections were taken and how they determined this location?

EM sections (images shown in Figures 1 and 3) were taken along the apical-basal axis through anterior lateral line neuromasts, particularly SO and IO neuromasts. The section orientation is best illustrated by Figure 1B, where the apical stereocilia are at the top of the image and basal is at the bottom. We added more detailed description of the section plane and specific neuromasts to the Results and Methods text, and indicated section plane in Figure 1A. Hair cells were identified based on their central location and darker cytoplasm as noted by (Owens et al., 2007; Behra et al., 2009; Suli et al., 2016). We do not see the stacked ER cistern morphology mentioned in (Chang et al., 1992). This is perhaps because lateral line hair cells more closely resemble type 2 hair cells which lack the perinuclear ER (Song et al., 1995; Nicolson, 2017). We thank the reviewers for drawing our attention to the difference in ER based on its proximity to the synaptic terminal. We re-analyzed our images using the post-synaptic density and synaptic ribbons as landmarks to help us identify pre-synaptic terminals. The data we included in the analysis presented in Figure 1 did not include the “thin near-membrane cistern” mentioned in (Fuchs, 2014).

2. Quantification of mitochondrial fragmentationIt is clear from the TEM cross sections that the mitochondria in hair cells (Figure 3 A) are quite different between pappaa mutants and controls. Whether there are mitochondria or ER networks are present is not apparent from these TEM images. Nor is it entirely clear that the networks are fragmented. The authors use plugins developed for confocal imaging to estimate fragmentation base on circularity and area/perimeter measurement. It is unclear if these measurement translate to hair cells or TEM. In addition to fragmentation in TEM images, the fragmented mitochondria in pappaa mutants are also hard to see in the live, max-projected mitoTimer images.The mitochondrial networks and fragmentation may be clearer or be better quantified by acquiring super resolution images of hair cells labeled with mitoTracker. In addition, it is possible that the fragmentation may also be visible or more convincing in videos of Z-stacks of mitoTracker label compared to in the max-projected images provided.

We would like to clarify that no plugin was used to analyze the EM images. We used the “free hand” tool to manually draw a ROI around each individual mitochondrion, then used the “analyze particles” function to measure the area, perimeter, and circularity. We understand that mitochondrial fragmentation is difficult to visualize in EM images. Therefore, we changed the text from “Pappaa loss causes mitochondrial fragmentation” to “Pappaa regulates mitochondrial morphology” which we believe is a better interpretation of our results. In addition, to better show the mitochondrial networks, we used Imaris software to make 3D reconstructions of the mitotracker-stained hair cells. In the revised manuscript Figure 3G, we include both the maximum intensity projection and sections in the xz and yz planes. We hope that the difference in mitochondrial morphology between wild type and *pappaa* mutants is more evident in these updated images.

3. Examination of hair cell ER morphologyThe previous work on Pappaa in zebrafish hair cells focused extensively on the mitochondria while the currently study the shifted the focus to the hair cell ER. While the ER-mito distances are convincing, a more wholistic picture of the amount or distribution of the ER in wildtype and mutant is lacking.This could be accomplished either using a transgenic line that labels the ER or a KDEL antibody (https://www.ncbi.nlm.nih.gov/pmc/articles/PMC4007406/).

We thank the reviewers for this suggestion. To analyze the amount of ER, we labeled the ER with KDEL antibody and measured the area occupied by KDEL immunolabeling, a quantification approach previously published by another zebrafish group (Blanco-Sánchez et al., 2014). We found that the percentage of neuromast area occupied by KDEL immunolabeling was similar between *pappaa* mutant and wild type hair cells, suggesting that the increase in ER-mitochondria associations is not caused by an expansion in ER. These new data are shown in Figure 1F and G.

4. It qualitatively appears that pappaa mutant hair cells are taking up a greater quantity of fluorescent Neo faster than WT i.e. the fluorescent intensity is greater in more hair cells. Did the authors quantify Neo-TR uptake?

We quantified Neo-TR fluorescence intensity by measuring the change in Neo-TR fluorescence over baseline (ΔF/F0) at 2.5, 4.5, and 6.5 minutes after exposure, and found no difference between genotypes suggesting the cells have similar rates of Neo-TR uptake. Additionally, we measured the maximum Neomycin-TR ΔF/F0 and found no difference between wild type and *pappaa* mutant hair cells, suggesting similar total amounts of neomycin are taken up by wild type and mutant hair cells. These new data are shown in Figure 4D and E.

5. Specificity of the pharmacological treatmentsThe authors perform numerous pharmacological experiments to disturb ER calcium. The authors suggest that their pharmacological manipulations trigger hair cell death due to the alteration in the interplay between ER/mito calcium in hair cells. What concentration of either of these drugs does it take to kill WT hair cells? Dose-response curves comparing WT and mutant would help support the idea that hair-cell death observed is a direct effect of the drugs on hair-cell ER-mitochondria calcium signaling.

We agree that dose-response curves provide a more reliable measure of drug effect. Given the high cost of Adensophostin A, we chose to focus on Thapsigargin experiments. In the revised manuscript, we include dose-response data for two ranges of Thapsigargin concentrations: lower concentrations with chronic exposure (Figure 2C) and higher concentrations with acute exposure (Figure 2D). Both these ranges span from doses that cause little to no hair cell death in wild type larvae to doses that cause low levels of wildtype cell death. At all doses, the mutants have lower hair cell survival rates than wild type.

Pharmacology is non-cell autonomous and the authors do not present evidence that these compounds specifically impact hair cell ER or mitochondrial calcium. Alternatively, these compound could impact supporting cell ER (https://elifesciences.org/articles/52160) as well as the ER in the innervating afferent or efferent neurons.More direct evidence show that hair cell mitochondria or ER calcium (measurements using mitoGCaMP such as in the previous study) are impacted by these treatments would make the author's claims more compelling.

We agree that we cannot rule out potential non-cell autonomous effects of Thapsigargin on neighboring cells. However, we specifically chose Thapsigargin because it was in fact previously shown to increase mitochondrial calcium levels (using mitoGCaMP fluorescence measurements) in zebrafish lateral line hair cells (Esterberg et al., 2014). We added the following text in the revised manuscript: “A previous study used a transgenic zebrafish line with a mitochondria-targeted calcium indicator (mitoGCaMP3) to show that Thapsigargin treatment caused increased mitochondrial calcium uptake in lateral line hair cells (Esterberg et al., 2014).”

6. The disconnect between IGFR1 and results in the current study.The identify and location of IGFR1 and the IGFBP are still undefined in this system and therefore it remains unclear exactly how IGRR1 or Pappaa impact sensory hair cells. In previous work on pappaa mutants (enhanced startle response, defects in photoreceptor synapse formation, defects in hair cell mitochondria) the role of IGFR1 in these processes was validated. In the current study, the link with IGFR1 is implied throughout.It is true that the relationship between IGFR1 and Pappa is well characterized and that currently the only known substrates of Pappa are IGFBPs. Despite this work, it is still possible that given the range of phenotypes in pappaa mutants, that Pappa has other protein substrates that have not yet been identified, or other has biological functions unrelated to the IGF system.To verify IGFR1 in this current study the authors could use NB1-31772 to stimulate IGF1 bioavailability and test whether this rescues either the autophagy or UPR defects in pappaa mutants. Being able to rescue these phenotypes also makes the study more compelling.

We include three new experiments in the revised manuscript to determine whether pappaa acts via IGF1 signaling to exert its effects on ER-calcium transfer and autophagy. First, we show that inducing expression of a dominant negative form of the IGF1 receptor increases hair cell sensitivity to Thapsigargin in *pappaa* mutants (Figure 2E). Second, we found that treatment with NBI-31772 to increase IGF1 bioavailability improves hair cell survival in *pappaa* mutant larvae treated with Thapsigargin (Figure 2F). Third, we show that treatment with NBI-31772 also rescues the autophagy defects in *pappaa* mutant hair cells (Figure 4F-G).

7. The authors state that there is not more spontaneous hair cell death in pappaa mutants compared to controls (line 443). Previous work has shown in zebrafish that Usher mutants (cdh23, ush1c, myo7a) also have an early UPR (https://www.ncbi.nlm.nih.gov/pmc/articles/PMC4007406/). Similar to pappaa mutants usher mutants have the same # of hair cells compared to controls, indicating no spontaneous hair cell death. But interestingly Usher mutants do have more TUNEL positive hair cells compared to controls, indicating that more hair cells in Usher mutants are in the process of apoptosis. Based on this new finding implicating the UPR response in pappaa mutants, could pappaa mutants, similar to hair cells in Usher mutants be more fragile (neomycin susceptible) as they are more likely to be in the process of apoptosis? A TUNEL label in pappaa mutants could reveal this. In addition, this paper on UPR in Usher mutant hair cells could be a useful paper to add to the discussion.

We thank the reviewers for this suggestion. We used TUNEL labeling and found that *pappaa* hair cells are not in the process of apoptosis. These new data are shown in Figure 5C. This result, together with our finding that the ER in *pappaa* mutant hair cells does not appear to undergo expansion (based on EM images and KDEL immunolabeling shown in Figure 1), which is indicative of more advanced ER stress, suggest that *pappaa* mutant hair cells experience less advanced ER stress compared to Usher mutants which do show ER expansion. We added text discussing the comparison between *pappaa* and Usher mutants to our discussion (page 8, lines 344-348)

8. Line 445-451: "Together, these findings suggest that Pappaa may regulate ER-mitochondria associations by promoting ER homeostasis. It is important to note that the ER and mitochondria are engaged in a constant feedback loop." This line of reasoning seems rather circular, considering that the previous study showed Pappaa regulates mitochondrial function. If mitochondrial function is impaired, it seems likely that ER homeostasis would be disrupted as well.

We modified the text in that paragraph to more clearly describe our reasoning and interpretation of our results. Although it is difficult to identify the primary defect when the ER-mitochondrial axis is disrupted, our work has further defined the cellular processes affected by loss of Pappaa. The new text (page 8-9, lines 355-373) reads as follows:

“Together, these findings suggest that Pappaa may regulate ER-mitochondria associations by promoting ER homeostasis. *pappaa^p170^* mutants display all the telltale signs of early ER stress, including activation of the adaptive UPR branch, heightened mitochondrial bioenergetics evident by increased calcium load, ROS, and hyperpolarization (Alassaf et al., 2019), and lack of spontaneous apoptosis (Figure 5C). […] Furthermore, given the novelty of Pappaa’s role in regulating ER-mitochondria associations, it will be interesting to identify the downstream molecular targets of Pappaa-mediated IGF1 signaling within the ER-mitochondria tethering complex.”

9. Methods: Overall, the methods section needs more detail. All experiments that were not previously performed by the author or the author's lab should have a concise description of what the authors did next to the reference (e.g. fish were imaged under Lab-Tek Chambered Coverglass (Fisher Scientific) where they were immobilized under a nylon mesh and two stainless-steel slice hold-downs (Warner Instruments) per Stawicki et al., 2014) A detailed description of how individual pappaa170 larvae used in experiments were genotyped is needed. A comprehensive description of how mitochondrial circularity was measured using the "mitochondrial morphology" plug-in in ImageJ is needed.

We revised our manuscript to add more detail throughout the methods section.

10. Statistics: how did the authors determine the power of the experiments were sufficient to avoid Type I and Type II error?

In general, we use the number of larvae as the N for statistical analyses, however, we analyzed at least 3 neuromasts per larva. In the revised manuscript, we include detailed numbers of neuromasts and larvae in the figure legends. All pharmacological experiments included a minimum of 24 neuromasts from 8 larvae per group which we believe to be sufficient to avoid type I and II error. For live imaging experiments, we were somewhat limited in the number of larvae we could test, because in order to maintain consistency of conditions among samples, we imaged all experimental and control groups for each experiment on the same day. The live imaging of Neo-TR autophagy experiment included 22 hair cells from 4 larvae per genotype, and the mitotracker experiment included 8 larvae per genotype. These numbers are in line with other zebrafish lateral line publications.

Alassaf M, Daykin EC, Mathiaparanam J, Wolman MA (2019) Pregnancy-associated plasma protein-aa supports hair cell survival by regulating mitochondrial function. *eLife* 8.

Behra M, Bradsher J, Sougrat R, Gallardo V, Allende ML, Burgess SM (2009) Phoenix is required for mechanosensory hair cell regeneration in the zebrafish lateral line. PLoS Genet 5:e1000455.

Blanco-Sánchez B, Clément A, Fierro J, Washbourne P, Westerfield M (2014) Complexes of Usher proteins preassemble at the endoplasmic reticulum and are required for trafficking and ER homeostasis. Dis Model Mech 7:547-559.

Chang JS, Popper AN, Saidel WM (1992) Heterogeneity of sensory hair cells in a fish ear. J Comp Neurol 324:621-640.

Esterberg R, Hailey DW, Rubel EW, Raible DW (2014) ER-mitochondrial calcium flow underlies vulnerability of mechanosensory hair cells to damage. J Neurosci 34:9703-9719.

Fuchs PA (2014) A 'calcium capacitor' shapes cholinergic inhibition of cochlear hair cells. J Physiol 592:3393-3401.

Kjaer-Sorensen K, Engholm DH, Kamei H, Morch MG, Kristensen AO, Zhou J, Conover CA, Duan C, Oxvig C (2013) Pregnancy-associated plasma protein A (PAPP-A) modulates the early developmental rate in zebrafish independently of its proteolytic activity. J Biol Chem 288:9982-9992.

Miller AH, Howe HB, Krause BM, Friedle SA, Banks MI, Perkins BD, Wolman MA (2018) Pregnancy-Associated Plasma Protein-aa Regulates Photoreceptor Synaptic Development to Mediate Visually Guided Behavior. J Neurosci 38:5220-5236.

Nicolson T (2017) The genetics of hair-cell function in zebrafish. J Neurogenet 31:102-112.

Owens KN, Cunningham DE, MacDonald G, Rubel EW, Raible DW, Pujol R (2007) Ultrastructural analysis of aminoglycoside-induced hair cell death in the zebrafish lateral line reveals an early mitochondrial response. J Comp Neurol 502:522-543.

Song J, Yan HY, Popper AN (1995) Damage and recovery of hair cells in fish canal (but not superficial) neuromasts after gentamicin exposure. Hear Res 91:63-71.

Suli A, Pujol R, Cunningham DE, Hailey DW, Prendergast A, Rubel EW, Raible DW (2016) Innervation regulates synaptic ribbons in lateral line mechanosensory hair cells. J Cell Sci 129:2250-2260.

Wilczak N, de Vos RA, De Keyser J (2003) Free insulin-like growth factor (IGF)-I and IGF binding proteins 2, 5, and 6 in spinal motor neurons in amyotrophic lateral sclerosis. Lancet 361:1007-1011.

Wolman MA, Jain RA, Marsden KC, Bell H, Skinner J, Hayer KE, Hogenesch JB, Granato M (2015) A genome-wide screen identifies PAPP-AA-mediated IGFR signaling as a novel regulator of habituation learning. Neuron 85:1200-1211.

[Editors' note: further revisions were suggested prior to acceptance, as described below.]

[…] Overall, this revision addressed a number of specific concerns experimentally, including providing pharmacological evidence that IGF system is involved in the hair cell processes they're examining. However some issues still need to be addressed:1. More is needed is needed on the locus of action where Pappaa/ IGF signaling regulates the hair cell ER-mitochondria axis. The first author has previously reported pappaa expression to be in supporting cells. However igf1rb expression in neuromasts has also been reported to be predominantly in supporting cells (e.g. https://piotrowskilab.shinyapps.io/neuromast_homeostasis_scrnaseq_2018/). If Pappa/IGF signaling is occurring in supporting cells, the effect on hair cells would be indirect. Resolution of where signaling occurs would be a valuable addition to this study. At the very least, this issue warrants discussion in the manuscript.

We added an experiment to ask where Pappaa/IGF signaling occurs, using an antibody against phosphorylated IGF1R (pIGF1R), which recognizes the IGF1R autophosphorylation event that occurs when the receptor is activated by IGF1 binding. This antibody has previously been validated using the IGF1R inhibitor NVP (Chablais and Jazwinska, 2010). We quantified the fluorescent intensity of pIGF1R in neuromasts and found that loss of Pappaa caused a reduction in pIGF1R fluorescence in both the hair cells and the support cells. The data are reported in Figure 2—figure supplement 1. The results and discussion of potential direct and indirect effects are added to the manuscript text on p. 6, lines 247-265. The detailed methods are reported in the Methods section p. 13, lines 521-528.

2. Currently the relationship between the numerous pathological issues present in pappaa hair cells (mitochondria and ER morphology disruption, lysosomal packaging) is unclear. Does the origin of the deficit rest in the mitochondria, the ER or another organelle? Overall, this work does reveal an additional mechanism (ER calcium) that renders loss of Pappaa detrimental to hair cell health. One caveat of the study is that it relies heavily on pharmacology which impacts not only the ER in hair cells, but also the glial-like supporting cells that surround hair cells, as well as the afferents and efferents that innervate this sensory system.These points would be worth discussion.

We have added more discussion of these caveats and questions to the manuscript text, in the Conclusion section, p. 9-10, lines 404-436.

3. The authors indicated in Figure 1A the orientation of the TEM sections and identified which anterior neuromasts. But it was not entirely clear how they determined the location of their organelles relative to neuronal contacts. There was no discussion in the manuscript on how they analyzed the images using the post-synaptic density and synaptic ribbons as landmarks. Additionally, there was no mention of efferent contacts in either the manuscript or the rebuttal.

We have added a new supplemental figure (Figure 1—figure supplement 1) showing images of the efferent contacts and synaptic ribbons in our EM sections. We added the following text to the manuscript, p. 4, lines 147-153:

“Because the ER at post-synaptic sites adjacent to efferent inputs has a distinct role in buffering high levels of post-synaptic calcium influx, and thus may impact the ER-mitochondria axis differently (Moglie et al., 2018), we did not include ER within 100 nm of the post-synaptic sites in our analysis. […] We also did not include ER in EM sections with presynaptic terminals, identified by synaptic ribbons (Figure 1—figure supplement 1B-B’).”

4. KDEL labeling does not appear to be specifically labeling ER in the images shown in Figure 1F. Instead, the labeling appears completely colocalized with hair cell plasma membrane bound GFP. For comparison, Blanco-Sánchez et al., 2014 showed discrete labeling of membrane within hair cells. Specificity of labeling should be verified if KDEL labeling will be used to measure the amount or distribution of the ER in wildtype and pappaa mutants.

The nuclei in the macula hair cells shown in the Blanco-Sanchez et al. study do not occupy as large a proportion of the cell as the nuclei in lateral line hair cells, thus allowing the ER in the cytoplasm to be spatially distinguished from the plasma membrane. In lateral line hair cells, the nucleus is large and the cytoplasm surrounding the nucleus is contained in a thin layer between the nucleus and plasma membrane. The previous image we showed in Figure 1F was of a single plane through the part of the cell containing the nucleus, and the resolution of our light microscopy was not sufficiently high to distinguish the thin layer of cytoplasm from the plasma membrane. We replaced that image with a maximum projection of several z-planes, including regions of the cytoplasm apical to the nucleus, where cytoplasmic ER labeling is distinguishable from the plasma membrane. The KDEL antibody was shown by Blanco-Sanchez (2018) to specifically label the ER by comparing to a transgenic line *Tg[myo6b:kdel-crimson]^b1319^* that labels the ER.

To complement the KDEL antibody labeling, we used another measure to assess the amount of ER present. We quantified the length of ER profiles in our EM sections, reasoning that increased abundance of ER in the cells may be reflected in longer tubule profiles. We found no difference in ER tubule profile length in mutants versus wild type hair cells. We also did not find a difference in the total number of ER tubules between mutants and wild type. We report these data in the new Figure 1H and have added the results to the manuscript text on p. 4, lines 169-173.

5. The authors did not answer the question of how they determined their experiments were sufficiently powered.

We used JMP Pro 15.0 software to perform power analyses for our experiments. Power analysis on the number of neuromasts, hair cells, or mitochondria we used for each experiment showed that all our experiments were sufficiently powered to avoid type 1 and 2 errors. We include this information in our Methods section, p. 16, lines 647-649, and added it to our Transparent Reporting form.

6. The x/z and y/z max intensity images of Mitotracker are better illustrations of what appear to be more fragmented mitochondria. The Imaris 3D reconstructions the authors mentioned in the rebuttal would have been informative, but they did not include them in Figure 3.

We now include videos of the 3D rendering of wild type and *pappaa* mutant mitotracker-labeled neuromasts (Video 1 and 2).

7. The P values for non-significance should be reported.

We included the p-values in the manuscript and the source data files

Blanco-Sánchez B, Clément A, Fierro J, Stednitz S, Phillips JB, Wegner J, Panlilio JM, Peirce JL, Washbourne P, Westerfield M (2018) Grxcr1 Promotes Hair Bundle Development by Destabilizing the Physical Interaction between Harmonin and Sans Usher Syndrome Proteins. Cell Rep 25:1281-1291.e1284.

Chablais F, Jazwinska A (2010) IGF signaling between blastema and wound epidermis is required for fin regeneration. Development 137:871-879.

Moglie MJ, Fuchs PA, Elgoyhen AB, Goutman JD (2018) Compartmentalization of antagonistic Ca. Proc Natl Acad Sci U S A 115:E2095-E2104.